# Heavy isotope labeling and mass spectrometry reveal unexpected remodeling of bacterial cell wall expansion in response to drugs

Heiner Atze[1], Yucheng Liang[1], Jean-Emmanuel Hugonnet[1], Arnaud Gutierrez[2], Filippo Rusconi[1,3*†], Michel Arthur[1*†]

[1]Centre de Recherche des Cordeliers, INSERM, Sorbonne Université, Université Paris Cité., Paris, France; [2]Robustesse et Evolvabilité de la vie, Institut Cochin, Université de Paris, INSERM U1016, CNRS UMR 8104, Paris, France; [3]PAPPSO, Université Paris-Saclay, INRAE, CNRS, Paris, France

*For correspondence:
filippo.rusconi@universite-paris-saclay.fr (FR);
michel.arthur@crc.jussieu.fr (MA)

†These authors contributed equally to this work

**Abstract** Antibiotics of the β-lactam (penicillin) family inactivate target enzymes called D,D-transpeptidases or penicillin-binding proteins (PBPs) that catalyze the last cross-linking step of peptidoglycan synthesis. The resulting net-like macromolecule is the essential component of bacterial cell walls that sustains the osmotic pressure of the cytoplasm. In *Escherichia coli*, bypass of PBPs by the YcbB L,D-transpeptidase leads to resistance to these drugs. We developed a new method based on heavy isotope labeling and mass spectrometry to elucidate PBP- and YcbB-mediated peptidoglycan polymerization. PBPs and YcbB similarly participated in single-strand insertion of glycan chains into the expanding bacterial side wall. This absence of any transpeptidase-specific signature suggests that the peptidoglycan expansion mode is determined by other components of polymerization complexes. YcbB did mediate β-lactam resistance by insertion of multiple strands that were exclusively cross-linked to existing tripeptide-containing acceptors. We propose that this undocumented mode of polymerization depends upon accumulation of linear glycan chains due to PBP inactivation, formation of tripeptides due to cleavage of existing cross-links by a β-lactam-insensitive endopeptidase, and concerted cross-linking by YcbB.

## Editor's evaluation

The authors describe the innovative use of a heavy-isotope labeling strategy combined with mass spectrometry analysis to investigate the role of peptidoglycan biosynthesis by an L,D-transpeptidase and penicillin-binding proteins in *Escherichia coli*. They use isotopic labelling of peptidoglycan followed by a chase experiment with labels to study how new subunits are assembled into the pre-existing sacculus. The data suggests that new material is inserted one strand at a time on the lateral wall while it appears to be inserted as multiple strands at the division septum. The data are novel and provide important insights, together with notable methodological advances.

## Introduction

The peptidoglycan is an essential component of the bacterial cell wall, which provides a mechanical barrier against the turgor pressure of the cytoplasm thereby preventing cells from bursting and lysing (*Vollmer and Bertsche, 2008*). The peptidoglycan also determines the shape of bacterial cells and is intimately integrated into the cell division process since the barrier to the osmotic pressure needs

**Figure 1.** Structure and biosynthesis of *Escherichia coli* peptidoglycan. (**A**) Structure of peptidoglycan. The net-like macromolecule is made of glycan strands cross-linked by short peptides. The polymerization of peptidoglycan involves glycosyltransferases that catalyze the elongation of glycan strands and transpeptidases that form 4→3 or 3→3 cross-links. (**B**) Structure of the peptidoglycan subunit. The stem peptide is assembled in the cytoplasm as a pentapeptide, L-Ala$^1$-D-iGlu$^2$-DAP$^3$-D-Ala$^4$-D-Ala$^5$, in which D-*iso*-glutamic acid (D-iGlu) and diaminopimelic acid (DAP) are connected by an amide bond between the γ-carboxyl of D-iGlu and the L stereo-center of DAP. (**C**) Formation of 4→3 cross-links. Active-site Ser transpeptidases belonging to the penicillin-binding protein (PBP) family catalyze the formation of 4→3 cross-links connecting the carbonyl of D-Ala at the 4th position of an acyl donor stem peptide to the side-chain amino group of DAP (D stereo-center) at the 3rd position of an acyl acceptor stem peptide. A pentapeptide stem is essential in the donor substrate to form the acyl enzyme intermediate. The acyl acceptor potentially harbors a tripeptide, a tetrapeptide, or a pentapeptide stem leading to the formation of Tetra-Tri, Tetra-Tetra, or Tetra-Penta dimers, respectively. (**D**) Formation of 3→3 cross-links. Active-site Cys transpeptidases belonging to the L,D-transpeptidase (LDT) family catalyze the formation of 3→3 cross-links connecting two DAP residues. LDTs are specific of tetrapeptide-containing donors.

The online version of this article includes the following figure supplement(s) for figure 1:

**Figure supplement 1.** Synthesis and recycling of peptidoglycan in *Escherichia coli*.

**Figure supplement 2.** Acceptor-to-donor radioactivity ratio (ADRR) as a function of the number of glycan strands inserted at the same time in the peptidoglycan layer.

**Figure supplement 3.** Peptidoglycan metabolism in *Escherichia coli* M1.5 (**A**) and Δ6*ldt* (**B**).

to be maintained during the entire cell cycle. These functions depend upon the net-like structure of the peptidoglycan macromolecule that is made of glycan strands cross-linked by short peptides (*Figure 1A*). It is assembled from disaccharide-peptide subunits consisting of β,1→4 linked *N*-acetyl-glucosamine (GlcNAc) and *N*-acetyl muramic acid (MurNAc) and a stem peptide linked to the D-lactoyl group of MurNAc (*Figure 1B*; *Mengin-Lecreulx et al., 1982*). Polymerization of the subunit is mediated by glycosyltransferases for the elongation of glycan chains and by transpeptidases for cross-linking stem peptides carried by adjacent glycan chains.

The last cross-linking step of peptidoglycan polymerization is catalyzed by two types of structurally unrelated transpeptidases (*Biarrotte-Sorin et al., 2006*), the L,D-transpeptidases (LDTs) (*Mainardi et al., 2008*) and the D,D-transpeptidases (*Zapun et al., 2008*), the latter belonging to the penicillin-binding protein (PBP) family. PBPs (*Figure 1C*) and LDTs (*Figure 1D*) use different acyl donor substrates (pentapeptide versus tetrapeptide) resulting in the formation of 4→3 versus 3→3 cross-links, respectively (*Hugonnet et al., 2016*; *Mainardi et al., 2005*). PBPs are inactivated by all classes of β-lactams, whereas LDTs are inactivated by carbapenems only (*Mainardi et al., 2007*). In wild-type *Escherichia coli*, LDTs are fully dispensable for growth, at least in laboratory conditions (*Magnet et al., 2008*). Accordingly, these enzymes have a minor contribution to peptidoglycan cross-linking in the exponential growth phase (*Vollmer and Bertsche, 2008*; *Glauner, 1988*). However, selection of mutants resistant to ceftriaxone, a β-lactam of the cephalosporin class, results in a full bypass of PBPs by the YcbB LDT (*Hugonnet et al., 2016*). The bypass requires overproduction of both the (p)ppGpp alarmone and YcbB. The peptidoglycan of the mutant grown in the presence of ceftriaxone only contains 3→3 cross-links following PBP inhibition.

D,D-carboxypeptidase (DDC) and L,D-carboxypeptidase activities sequentially remove D-Ala residues at the 5th and 4th positions of stem peptides containing a free carboxyl end, which are present in the acceptor position of dimers and in monomers. DDCs, which belong to the PBP family, hydrolyze the D-Ala$^4$-D-Ala$^5$ amide bond of stem pentapeptides to generate stem tetrapeptides in mature peptidoglycan (*Sauvage et al., 2008*). Since the reaction is nearly total in *E. coli*, stem pentapeptides are found in very low abundance unless DDCs are inactivated by β-lactams. The L,D-carboxypeptidase activity of LDTs hydrolyzes the DAP$^3$-D-Ala$^4$ amide bond generating tripeptide stems from tetrapeptide stems (*Magnet et al., 2008*). Since this reaction is incomplete the peptidoglycan contains combinations of tripeptide and tetrapeptide stems in monomers and in the acyl acceptor position of dimers. Removal of D-Ala residues by DDCs and LDTs occurs before or after cross-linking, leading to the polymorphism in the acceptor position of dimers depicted in *Figure 1C and D*.

By removing D-Ala$^5$ from pentapeptide stems, DDCs eliminate the essential pentapeptide donor of PBPs and generate the essential tetrapeptide donor of LDTs (*Mainardi et al., 2002*; *Hugonnet et al., 2016*). By removing D-Ala$^4$ from tetrapeptide stems, LDCs eliminate the essential tetrapeptide donor of LDTs thereby potentially limiting the formation of 3→3 cross-links. Thus, DDC and LDC activities control both the extent of peptidoglycan cross-linking and the relative contributions of D,D-transpeptidases and LDTs to peptidoglycan cross-linking.

The interconversion between monomers and cross-linked dimers involves not only PBPs and LDTs in the biosynthetic direction but also hydrolytic enzymes, referred to as endopeptidases, which cleave the cross-links. Endopeptidases belonging to the PBP family are specific of 4→3 cross-links whereas other endopeptidases cleave both 4→3 and 3→3 cross-links (*Voedts et al., 2021*). Among eight endopeptidases with partially redundant functions, at least one enzyme is essential in the context of the formation of 4→3 cross-links whereas two endopeptidases, MepM and MepK, are required in the context of 3→3 cross-links (*Voedts et al., 2021*; *Singh et al., 2012*).

Morphogenesis of *E. coli* cells depends upon controlled expansion of the peptidoglycan and requires scaffolding proteins, actin (MreB) and tubulin (FtsZ) homologues (*den Blaauwen et al., 2008*). These proteins are involved in the recruitment of peptidoglycan polymerases and in the coordination of their activities with that of hydrolases (*Vollmer and Höltje, 2001*). The cell cycle involves two phases, namely the MreB-dependent elongation of the side wall at a constant diameter and the FtsZ-dependent formation of the septum at mid-cell. These processes are thought to involve two distinct multienzyme complexes, the elongasome and the divisome, and specific enzymes, in particular class B PBP2 and PBP3, which mediate peptidoglycan cross-linking in the side wall and in the septum, respectively (*Höltje, 1996*). Accordingly, specific inactivation of PBP2 by mecillinam or of PBP3 by aztreonam results in the growth of *E. coli* as spheres or as filaments, respectively.

The average chemical composition of peptidoglycan is well known. Most enzymes involved in the formation and cleavage of glycosidic and amide bonds have been identified (at least one enzyme and generally several enzymes for each bond). However, the spatial arrangement of glycan strands is still a matter of debate. Most models propose a regular arrangement of glycan strands parallel to the cell surface and perpendicular to the long axis of the cell as depicted in *Figure 1A*; *Typas et al., 2011*. Alternatively, it has been proposed that the glycan strands protrude perpendicularly to the surface of the cytoplasmic membrane (*Meroueh et al., 2006*; *Dmitriev et al., 2003*). The lack of direct experimental evidence originates from the heterogeneity of the peptidoglycan macromolecule that prevents the analysis of the polymer by radiocrystallography. NMR spectroscopy also failed to determine the precise and complete structure of the polymer due to its heterogeneity and its important flexibility, as measured by solid-state NMR relaxation (*Kern et al., 2008*; *Sharif et al., 2009*). The mode of insertion of new glycan strands in the growing peptidoglycan is also not fully characterized. According to the 'three-for-one' model, peptidoglycan expansion involves the insertion of three neosynthesized glycan strands to the detriment of the hydrolysis of one existing strand, which acts as a docking strand (*Vollmer and Höltje, 2001*). Alternatively, glycan strands may be inserted according either to a 'one-at-a-time' model or a 'two-at-a-time' model, the latter being supported by the dimeric nature of certain D,D-transpeptidases (*Charpentier et al., 2002*). All these models postulate that peptidoglycan expansion requires cleavage of cross-links by endopeptidases and obey to the 'make-before-break' principle, which proposes that cross-links in the existing peptidoglycan are hydrolyzed by endopeptidases only if the newly synthesized glycan strands are cross-linked and can sustain the osmotic pressure of the cytoplasm (*Vollmer and Höltje, 2001*).

At each generation, about half of the disaccharide-peptide units is released from the peptidoglycan by the combined action of endopeptidases and lytic transglycosylases (*Goodell, 1985a*; *Johnson et al., 2013*). The latter enzymes catalyze a non-hydrolytic cleavage of the β,1→4 MurNAc-GlcNAc glycosydic bond and generate GlcNAc-anhydro-MurNAc-peptides fragments via MurNAc cyclisation at positions 1 and 6. These fragments are transported into the cytoplasm by the AmpG permease and recycled according to the pathway depicted in *Figure 1—figure supplement 1*. The key features of the recycling pathway are that the L-Ala-γ-D-Glu-DAP tripeptide is directly recycled whereas the glucosamine moiety of GlcNAc and MurNAc reenters into the peptidoglycan biosynthesis pathway at the level of glucosamine-6-phosphate. The relationships between peptidoglycan synthesis and recycling are poorly understood. The 'three-for-one' growth model predicts that the docking strand is recycled; conversely, the 'one-at-a-time' and 'two-at-a-time' models do not imply that recycling is a consequence of peptidoglycan synthesis.

The peptidoglycan expansion models described above were previously investigated by pulse labeling of peptidoglycan with radioactive DAP. Indeed, the models imply specific acceptor-to-donor radioactivity ratios (ADRR) in cross-linked stem peptides (*Figure 1—figure supplement 2*; *de Jonge et al., 1989*). The ADRR may potentially vary from zero to infinite for (i) exclusive cross-linking of neo-synthesized (radioactive) donor stems to existing (unlabeled) acceptor stems (new→old; ADRR=0), (ii) the formation of cross-links between neo-synthesized (radioactive) donor and acceptor stems (new→new; ADRR=1), and (iii) cross-linking of existing (unlabeled) donor stems to neo-synthesized (labeled) acceptor stems (old→new; infinite ADRR). This method is intrinsically inaccurate since newly synthesized subunits issued from the recycling of stem peptides are mistakenly identified as material from the existing wall. Moreover, the method provides access to the average composition of dimers in labeled donor and acceptor stems rather than the labeling status of stem peptides that have directly participated in the same cross-linking reaction. Thus, an ADRR of 1 can originate either from dimers containing neo-synthesized stem peptides in the donor and acceptor positions (new→new) or, alternatively, from a combination of dimers containing a radioactive DAP either in the donor (new→old) or acceptor (old→new) position in equimolar amounts. This limitation was disregarded in previous studies since dimers containing a single labeled DAP residue in the donor position were considered to be rare, at least in wild-type *E. coli* grown in the absence of β-lactam. This is arguably the case for dimers generated by PBPs since pentapeptide stems are rapidly converted to tetrapeptide stems by DDCs thereby precluding existing stems from participating as donors in the formation of 4→3 cross-links by D,D-transpeptidases. Obviously, this premise does not apply to LDTs since these enzymes use tetrapeptide stems as donors (*Figure 1*). Here, we describe a new method based on the full labeling of the entire peptidoglycan macromolecule with stable isotopes of carbon and nitrogen combined with high-resolution mass spectrometry (MS). This method does not suffer from the limitations listed above and our results show that it enables a thorough investigation of peptidoglycan synthesis and recycling in a single experiment. We applied our method to the comparison of the modes of expansion of the peptidoglycan in bacterial strains relying on PBPs, and, for the first time, on LDT YcbB, or on both types of enzymes for peptidoglycan cross-linking. We show how previously described β-lactam-induced malfunctioning of the bacterial cell wall synthesis machinery (*Cho et al., 2014*) affects peptidoglycan synthesis and recycling. We also report how LDT YcbB rescues inhibition of PBPs by participating in an unprecedented mode of peptidoglycan polymerization. The results described in this report were obtained, thanks to the richness of the structural information brought by MS and MS/MS analyses of all the labeled/unlabeled sugar and amino acid residues of peptidoglycan subunits, as opposed to the radioactive signal located in the DAP moiety only. The mass data analysis pipeline automating the identification of labeled muropeptides is described in this study and may be generally applied to the screening of the mode of action of antibacterial agents acting on peptidoglycan synthesis.

## Results

### Muropeptide composition of the peptidoglycan from *E. coli* M1.5

Strain M1.5 was chosen for initial experiments because D,D-transpeptidases and LDTs both have major contributions to peptidoglycan polymerization offering the possibility to study 4→3 and 3→3 cross-linked dimers from the same peptidoglycan preparation. Strain M1.5 was grown in labeled

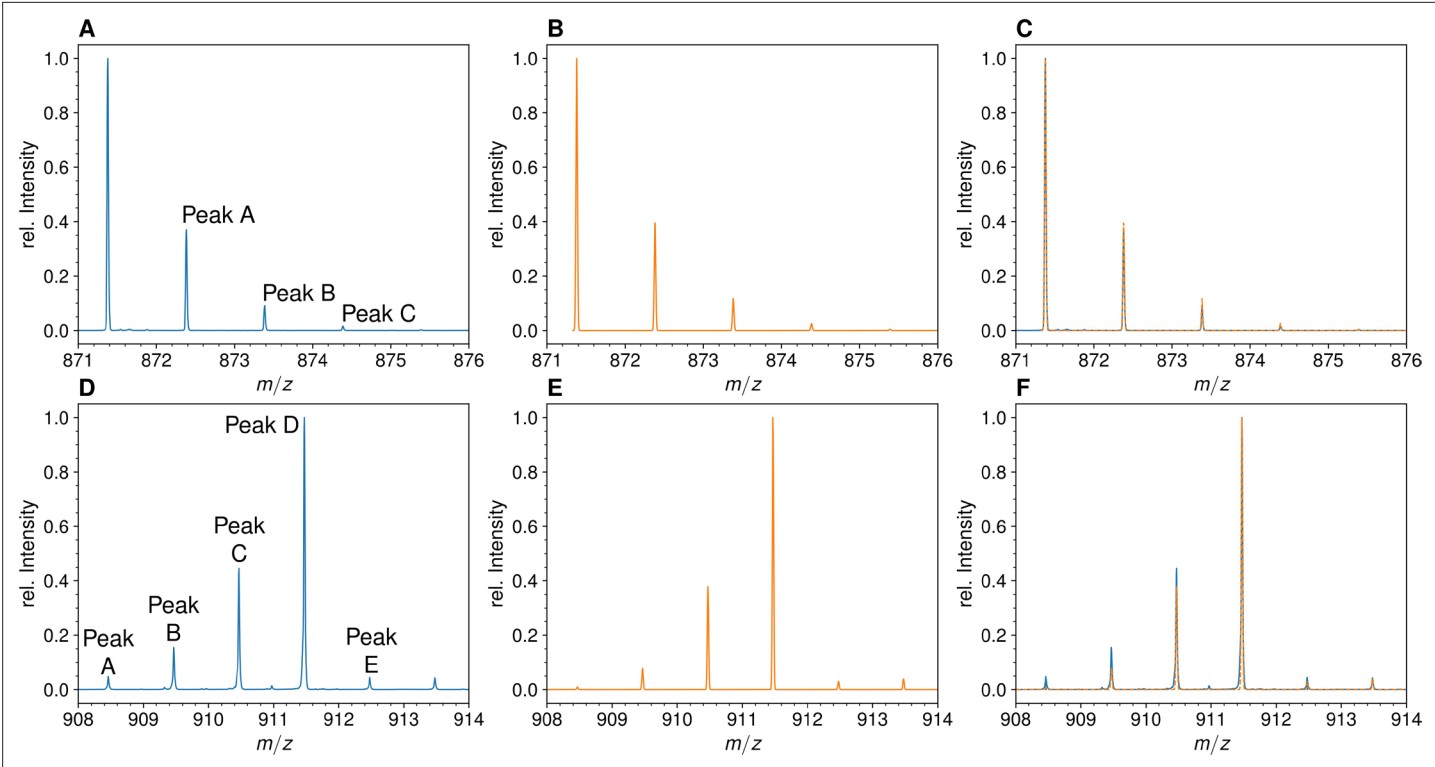

**Figure 2.** Mass spectra of the unlabeled and fully labeled disaccharide-tripeptide monomers. (**A**) Experimental mass spectrum of the mono-protonated ([M+H]⁺) disaccharide-tripeptide ($C_{34}H_{58}N_6O_{20} \bullet H^+$) extracted from *Escherichia coli* M1.5 grown in unlabeled M9 minimal medium. (**B**) Simulated spectrum obtained for the natural abundance of carbon and nitrogen isotopes. (**C**) Overlay of the spectra in A and B. (**D**) Observed mass spectrum of the GlcNAc-MurNAc-tripeptide purified from the peptidoglycan of *E. coli* M1.5 grown in the labeled M9 minimal medium. (**E**) Simulated mass spectrum obtained for 99% ¹³C and ¹⁵N labeling. (**F**) Overlay of spectra in D and E. GlcNAc, *N*-acetylglucosamine; MurNAc, *N*-acetyl muramic acid.

The online version of this article includes the following figure supplement(s) for figure 2:

**Figure supplement 1.** Mass spectra of the unlabeled and fully labeled disaccharide-tetrapeptide monomers.

**Figure supplement 2.** Mass spectra of unlabeled and fully labeled Tetra→Tetra dimer.

M9 minimal medium containing [¹⁵N]NH₄Cl and [¹³C]glucose as the sole sources of nitrogen and carbon, respectively. *E. coli* M1.5 was also grown in unlabeled M9 minimal medium. Peptidoglycan was extracted from exponentially growing cultures and digested by muramidases, which generate soluble disaccharide-peptides. The resulting muropeptides were reduced with NaBH₄ and purified by *rp*HPLC. MS analyses (see below) were performed to determine the structure of the six predominant muropeptides. These muropeptides included two monomers containing either a tripeptide (L-Ala¹-D-iGlu²-DAP³) or a tetrapeptide (L-Ala¹-D-iGlu²-DAP³-D-Ala⁴) stem, both linked to a reduced GlcNAc-MurNAc disaccharide. The remaining four muropeptides were dimers containing a 3→3 or a 4→3 cross-link with a tripeptide or a tetrapeptide stem in the acceptor position in all four combinations. As expected, growth in the labeled versus unlabeled medium did not alter the muropeptide composition of peptidoglycan. The average abundance of the six predominant muropeptides is reported in *Supplementary file 1a*.

## Determination of the isotopic composition of the unlabeled monomers

The mass spectra of monomers from *E. coli* grown in unlabeled M9 minimal medium displayed the conventional isotopic clusters (*Figure 2*, *Figure 2—figure supplement 1* for the disaccharide-tripeptide and -tetrapeptide, respectively), as expected for the natural abundances of carbon, hydrogen, nitrogen, and oxygen isotopes (see Supplementary Materials and methods for the method used to deduce the mass and abundance of isotopologues from the isotopic composition of the medium). For the disaccharide-tripeptide, the most abundant isotopologue (64.382%) only contained the light isotopes of the four chemical elements. The second peak of the isotopic cluster, peak A, was

composed of isotopologues containing only light isotopes except for either one $^{13}$C (23.497%), $^{2}$H (0.437%), $^{15}$N (1.411%), or $^{17}$O (0.492%) isotope, together representing ca. 26% of the relative abundance of the most abundant isotopologues. The overlay of the experimental and simulated spectra (*Figure 2C*) confirmed the expected isotopologue composition (see Supplementary Materials and methods for the method used to simulate mass spectra by using the deduced mass and abundance of isotopologues).

## Determination of the isotopic composition of the fully labeled monomers

MS analyses were performed on the reduced disaccharide-tripeptide monomer purified from the peptidoglycan of strain M1.5 grown in the labeled medium (*Figure 2D*). The theoretical mass spectra were simulated as described in the Supplementary Materials and methods section (*Figure 2E*) and overlaid to the experimental spectra (*Figure 2F*), revealing a good match between the two types of spectra. For the labeled tripeptide, the most intense peak of the isotopic cluster at *m/z* 911.473 (*z*=1) (*Figure 2D*) corresponds to peak D described in the Supplementary Materials and methods, which is mainly accounted for by the uniformly labeled disaccharide-tripeptide (*m/z* value of 911.474). Additional isotopic cluster peaks at *m/z* values of 908.463, 909.467, and 910.469 (*Figure 2D*), correspond to peaks A, B, and C (*m/z* values of 908.467, 909.469, and 910.472). These peaks correspond to uniformly labeled species except for the presence of combinations of 3, 2, and 1 light $^{12}$C or $^{14}$N nuclei, respectively. The presence of these peaks fits the simulated spectra that were calculated with the extent of [$^{15}$N]NH$_4$Cl and [$^{13}$C]glucose labeling reported by the manufacturers (99% for each isotope; please note that both the preculture and the culture were performed in labeled medium to avoid isotopic dilution). The same conclusion was drawn from the comparison of the simulated and experimental spectra of the labeled disaccharide-tetrapeptide (*Figure 2—figure supplement 1*). Qualitatively, it is worth noting that the mass spectra of the unlabeled and labeled disaccharide-tripeptide presented in *Figure 2* are almost mirror images. This reflects the preponderance of the isotopologue containing only light isotopes in the unlabeled muropeptide followed (to the right) by peaks of decreasing intensity resulting from incorporation of a few rare heavy isotopes. In contrast, the peak of the fully labeled disaccharide-peptide is preceded (to the left) by peaks of decreasing masses due

**Table 1.** Tandem mass spectrometry analysis of isotopologues of the GlcNAc-MurNAc-tetrapeptide.

| Structure | Isotopologue | $m/z_{cal}$ | Fully labeled | | One light nucleus | |
|---|---|---|---|---|---|---|
| | | | $m/z_{obs}$ | Intensity (%) | $m/z_{obs}$ | Intensity (%) |
| GlcNAc-MurNAc-L-Ala-γ-D-Glu-DAP-D-Ala* | Fully labeled | 986.52 | 986.52 | 100 | ND | 0 |
| | One light nucleus | 985.52 | ND | 0 | 985.60 | 100 |
| MurNAc-L-Ala-γ-D-Glu-DAP-D-Ala | Fully labeled | 774.42 | 774.42 | 100 | 774.45 | 18 |
| | One light nucleus | 773.41 | ND | 0 | 773.45 | 82 |
| MurNAc-L-Ala-γ-D-Glu-DAP | Fully labeled | 681.36 | 681.36 | 100 | 681.40 | 12 |
| | One light nucleus | 680.36 | ND | 0 | 680.40 | 88 |
| L-Ala-γ-D-Glu-DAP-D-Ala | Fully labeled | 485.27 | 485.27 | 100 | 485.30 | 35 |
| | One light nucleus | 484.26 | ND | 0 | 484.30 | 65 |
| γ-D-Glu-DAP-D-Ala | Fully labeled | 410.22 | 410.22 | 100 | 410.25 | 47 |
| | One light nucleus | 409.22 | ND | 0 | 409.25 | 53 |
| L-Ala-γ-D-Glu-DAP | Fully labeled | 392.21 | 392.21 | 100 | 392.25 | 48 |
| | One light nucleus | 391.21 | ND | 0 | 391.25 | 52 |
| γ-D-Glu-DAP | Fully labeled | 317.17 | 317.17 | 100 | 317.20 | 54 |
| | One light nucleus | 316.16 | ND | 0 | 316.20 | 46 |

*Mono-protonated precursor ion: $C_{37}H_{63}N_7O_{21}\bullet H^+$.

$m/z_{cal}$ = calculated *m/z*. $m/z_{obs}$ = observed *m/z*; ND = not detected.

to incorporation of a few rare $^{12}$C and $^{14}$N nuclei. The same conclusions apply to the spectrum of the unlabeled disaccharide-tetrapeptide monomer (*Figure 2—figure supplement 1*).

The structure of the isotopologues was determined by MS/MS. As an example, *Table 1* presents the MS/MS data obtained for mono-protonated isotopologue ions of the labeled disaccharide-tetrapeptide ($C_{37}H_{63}N_7O_{21} \bullet H^+$) under isotopic cluster peaks at $m/z_{obs}$ 986.519 and 985.516. These values match the calculated $m/z_{cal}$ values for isotopologues either exclusively containing the $^{13}$C and $^{15}$N isotopes ($m/z$ 986.518) or bearing a single light isotope of either element ($m/z_{cal}$ 985.515 or 985.521 for a $^{12}$C or $^{14}$N isotope, respectively). Fragmentation of the isotopologues under the peak at $m/z_{obs}$ of 986.519 resulted in the expected set of product ions for a uniformly labeled disaccharide-tetrapeptide, while two sets of product ions, differing by approximately one mass unit, were observed for the fragmentation of isotopologues bearing a single light isotope under the peak at $m/z_{obs}$ of 985.516. The fragments in the lower-mass ion set had retained their single light nucleus ($^{12}$C or $^{14}$N) while the fragments in the higher-mass ion set had lost their single light nucleus and were fully labeled (these higher-mass fragments matched exactly those obtained for the fully labeled disaccharide-tetrapeptide). Thus, each fragment was present in the spectrum as a pair of peaks that reflected the presence or the absence of the light isotope. *Table 1* provides the relative intensity of these two peaks for each fragment of the isotopologues. The relative intensity of the peak containing the light isotope decreased with the mass of the fragment. This result is expected for a uniform distribution of the light isotope in both the sugar and peptide moieties of the disaccharide-tetrapeptide because a smaller number of nuclei (smaller mass) makes it more probable to lose the single light nucleus during fragmentation.

Taken together, these results show that we successfully achieved extensive peptidoglycan labeling that was only limited by the extent (99%) of the labeling of glucose and $NH_4Cl$ introduced in the labeled growth medium. Thus, isotopic dilution due to atmospheric $N_2$ and $CO_2$ was not an issue, as expected from the metabolic fluxes of *E. coli* grown in minimal medium. Note that the preculture and the culture were both performed in labeled M9 medium to avoid any contribution of the initial inoculum to the final isotopic composition of the peptidoglycan. Simulation of high-resolution mass spectra was successfully used to assign isotopologues of defined structure and isotopic composition to the corresponding peaks in experimental mass spectra (*Figure 2*, *Figure 2—figure supplement 1*). These assignments were confirmed by MS/MS (shown for two isotopologues in *Table 1*).

## Mass spectra of the four dimers

Peptidoglycan dimers do not harbor any symmetry axis, as one peptide stem acts as an acyl donor in the transpeptidation reaction whereas the other acts as an acyl acceptor and retains the single free carboxyl extremity of the peptide moiety (*Figure 1*). By convention, the structure of dimers is represented by placing the donor on the left separated from the acceptor by an arrow (donor→acceptor). In *E. coli* M1.5, the YcbB LDT catalyzes the formation of DAP$^3$→DAP$^3$ cross-links connecting two diaminopimelyl residues (DAP$^3$) located at the third position of a donor stem (a tripeptide) and of an acceptor stem (a tripeptide or a tetrapeptide), thus generating Tri→Tri and Tri→Tetra dimers, respectively. These dimers only differ by the presence or absence of D-Ala at the C-terminal end of the acceptor stem. The D,D-transpeptidases of *E. coli* M1.5 catalyze the formation of D-Ala$^4$→DAP$^3$ cross-links connecting a tetrapeptide donor stem to an acceptor stem containing either a tripeptide (Tetra→Tri dimer) or a tetrapeptide (Tetra→Tetra dimer). In these dimers, D-Ala at the fourth position of the donor stem is engaged in the D-Ala$^4$→DAP$^3$ cross-link and DAP$^3$ or D-Ala$^4$ occupies the C-terminal position of the acceptor stem. The Tri→Tetra and Tetra→Tri isomers formed by YcbB and the D,D-transpeptidases can be discriminated by MS/MS on the basis of the cleavage of the DAP$^3$→DAP$^3$ and D-Ala$^4$→DAP$^3$ cross-links, respectively, and of the loss of a C-terminal D-Ala only present in the acceptor stem of the dimer generated by YcbB (Supplementary data). As detailed above for monomers, the comparison of the experimental and simulated mass spectra showed that the observed isotopic clusters of the dimers can be accounted for by the isotopic composition of the growth media (see *Figure 2—figure supplement 2* for the Tetra→Tetra dimer). The structure of all dimers was confirmed by MS/MS (*Supplementary file 2*). The peptidoglycan preparations also contained small amounts of trimers and tetramers that were not analyzed in the current study because their low abundance, poor resolution by *rp*HPLC, and complex isotopomer composition precluded their full characterization by MS and MS/MS.

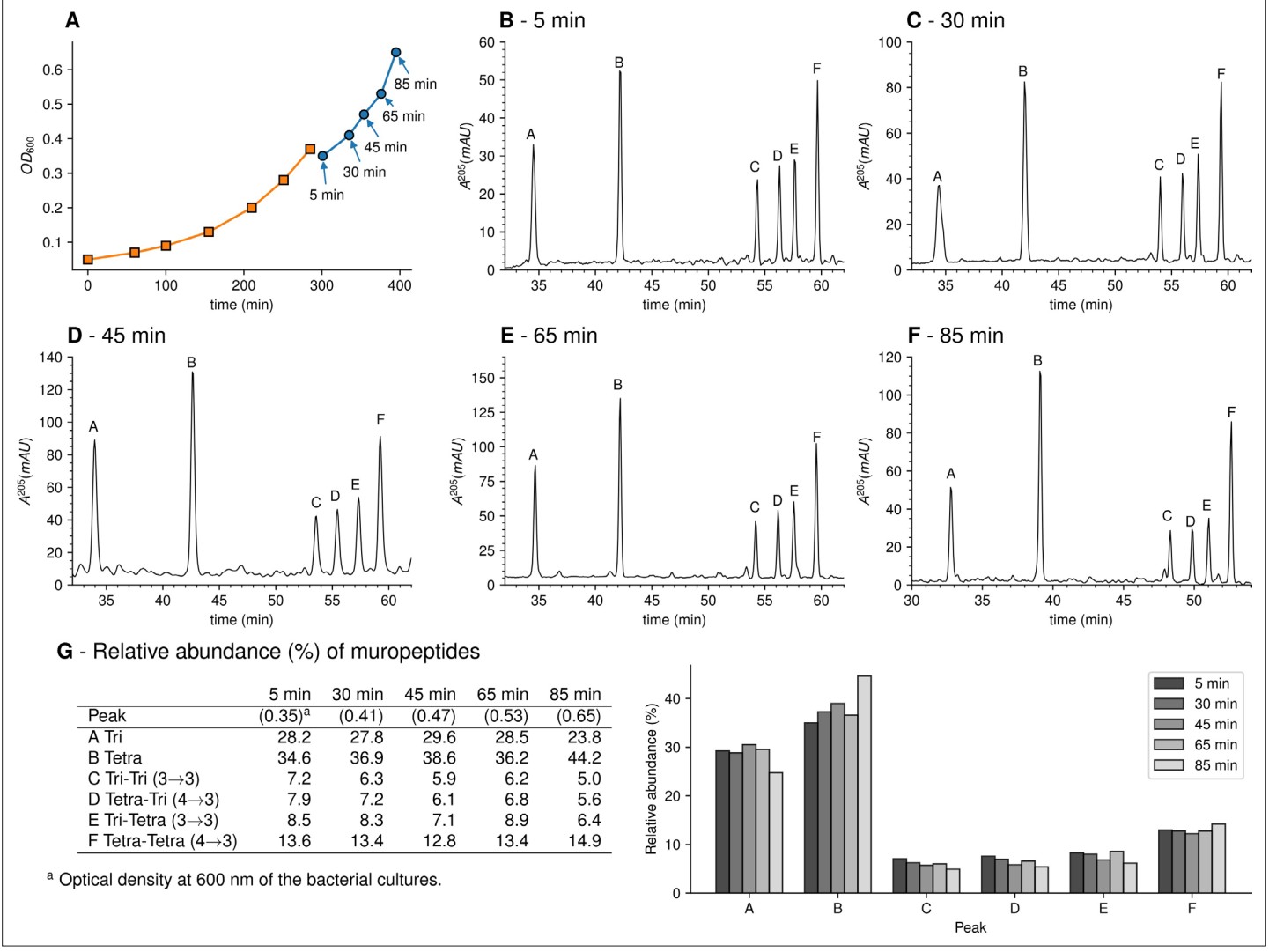

**Figure 3.** Muropeptide profile of peptidoglycan extracted from *Escherichia coli* M1.5 after the medium switch. (**A**) *E. coli* M1.5 was inoculated in the labeled minimal medium and bacterial growth was monitored by determining the $OD_{600}$ (orange curve, squares). When the optical density reached 0.4, bacteria were collected by centrifugation, resuspended in the unlabeled minimum medium, incubation was continued (blue curve), and samples were withdrawn at 5, 30, 45, 65, and 85 min (circles). (**B**) to (**F**) Peptidoglycan was extracted from these five culture samples, digested with muramidases, and the resulting muropeptides were separated by *rp*HPLC. (**G**) Identification and relative abundance of muropeptides in the major chromatographic peaks.

## Study design for the kinetics of incorporation of unlabeled subunits into the fully labeled peptidoglycan of strain M1.5

To investigate the mode of insertion of peptidoglycan precursors into sacculi, *E. coli* M1.5 was grown in the heavy isotope labeled minimal medium to an $OD_{600}$ of 0.4. Bacteria were collected by centrifugation and resuspended in the unlabeled medium. Incubation was continued in the same conditions and samples were collected at 5, 30, 45, 65, and 85 min after the medium switch (*Figure 3A*), representing a little less than a generation (90 min; *Supplementary file 1b*). Peptidoglycan was extracted and the six predominant muropeptides, two monomers and four dimers, were purified by *rp*HPLC (*Figure 3B-F*) and identified by MS (*Figure 3G*). The relative abundance of these six muropeptides remained similar in all samples indicating, as expected, that the medium switch did not alter the overall peptidoglycan composition.

## Impact of the medium switch on the isotopic composition of monomers

MS revealed that growth in the unlabeled medium after the switch led to the appearance of disaccharide-tripeptide isotopologues of natural isotopic composition (*Figure 4A*). These isotopologues were

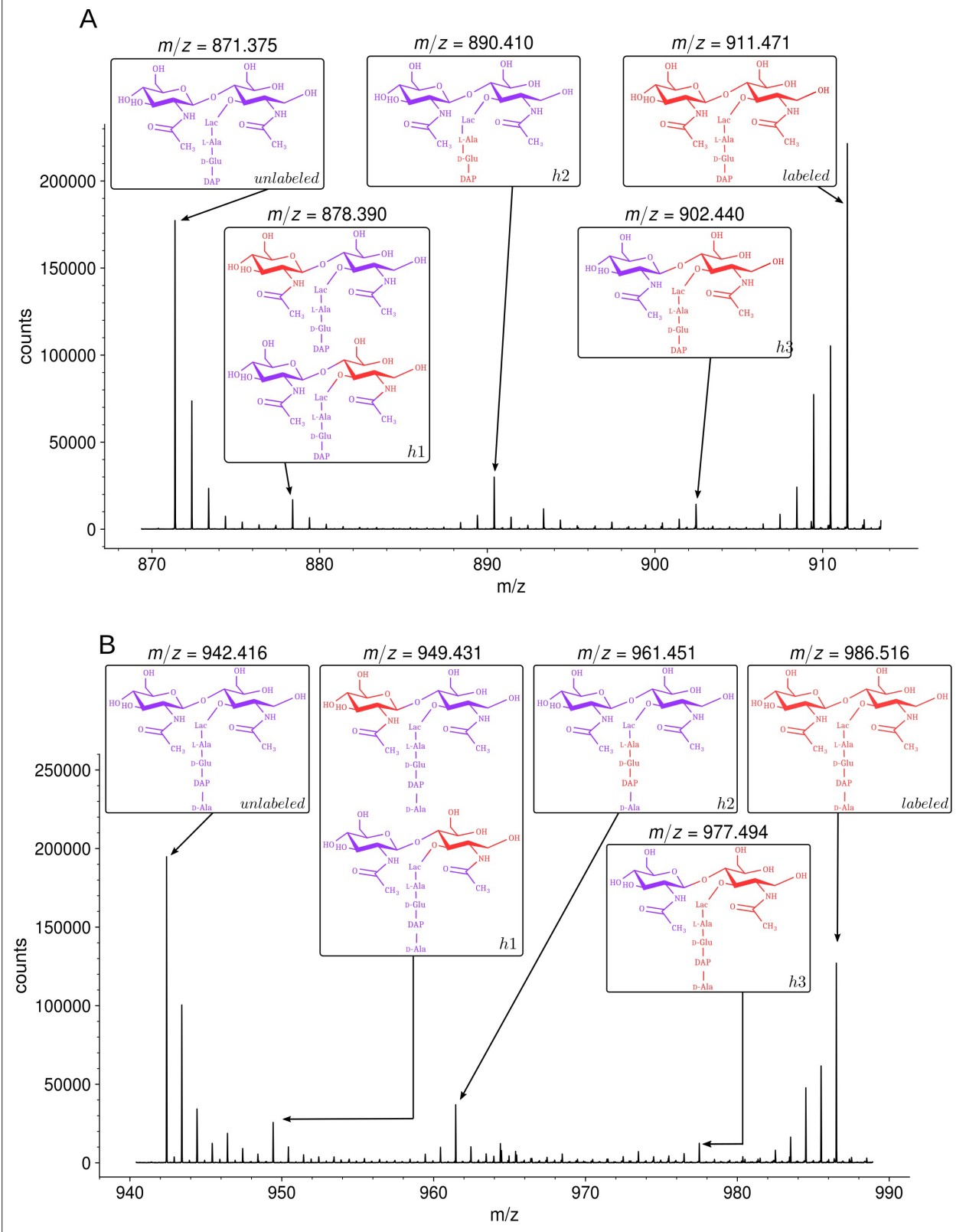

**Figure 4.** Isotopic composition of monomers isolated from the peptidoglycan of *Escherichia coli* M1.5 after the medium switch. (**A**) Mass spectrum obtained for the GlcNAc-MurNAc-tripeptide monomer recovered in peak A of the chromatogram in *Figure 3D*. (**B**) Mass spectrum obtained for the GlcNAc-MurNAc-tetrapeptide monomer recovered in peak B of the chromatogram in *Figure 3D*. Purple and red colors indicate the unlabeled ($^{12}$C

*Figure 4 continued on next page*

*Figure 4 continued*

and $^{14}$N) and labeled ($^{13}$C and $^{15}$N) isotopic content of the amino acid and sugar residues of the monomers, respectively. GlcNAc, *N*-acetylglucosamine; MurNAc, *N*-acetyl muramic acid.

The online version of this article includes the following figure supplement(s) for figure 4:

**Figure supplement 1.** Isotopic composition of monomers isolated from the peptidoglycan of *Escherichia coli* M1.5 and a recycling-deficient derivative obtained by deletion of the ampG permease gene.

**Figure supplement 2.** Simulated spectra of hybrid monomers.

exclusively assembled from unlabeled glucose and NH$_4$Cl after the medium switch. In addition, minor amounts of hybrids containing both labeled and unlabeled moieties were detected (named h1, h2, and h3 in *Figure 4A*). MS/MS (*Supplementary file 2*) indicated that hybrid h1 consists of two types of isotopologues containing a labeled hexosamine moiety either in the GlcNAc or MurNAc residue. Hybrid h2 contained a labeled stem peptide and an unlabeled disaccharide. Hybrid h3 was uniformly labeled except for GlcNAc. Hybrids h1, h2, and h3 could originate from the incorporation of recycled fragments of existing (labeled) peptidoglycan. This requires digestion of the existing peptidoglycan by hydrolases and import of the digested fragments by specialized permeases (AmpG and Opp) (*Johnson et al., 2013*). Alternatively, the hybrids could originate from an intracellular pool of labeled precursors present prior to the medium switch. The structures of h1 and h2 were fully accounted for by the known peptidoglycan recycling pathway of *E. coli* (*Johnson et al., 2013*; *Figure 1—figure supplement 1*). Indeed, GlcNAc and MurNAc are recycled into glucosamine-6-phosphate, a common precursor of UDP-GlcNAc and UDP-MurNAc. This pathway could therefore account for the incorporation of a recycled (labeled) hexosamine into the GlcNAc or MurNAc residues of the h1 isotopologues. Regarding h2, the peptide moiety of peptidoglycan fragments is recycled as a tripeptide, which is added to UDP-MurNAc by the Mpl synthase (*Mengin-Lecreulx et al., 1996*). This pathway could be responsible for the formation of the h2 hybrid containing a labeled (recycled) tripeptide and an unlabeled GlcNAc-MurNAc disaccharide. In contrast, the formation of the h3 hybrid did not depend upon recycling since MurNAc-peptide moieties are not selectively recycled as single molecules. Thus, the h3 hybrid originated from moieties that were assembled after (unlabeled GlcNAc) or prior to (labeled MurNAc-peptide) the medium switch. Our results are consistent with the fact that the cytoplasm of *E. coli* contains an important pool of UDP-MurNAc-pentapeptide (an estimate of 120,000 molecules per bacterial cell), which represents ca. 3% of the total number of disaccharide-peptide units present in the cell wall (*Mengin-Lecreulx et al., 1982*). These observations indicate that h3 is the sole hybrid to be assembled independently from recycling.

The way recycling yields different kinds of isotopologues was further investigated by constructing a derivative of *E. coli* M1.5 deficient in peptidoglycan recycling following deletion of the *ampG* permease gene. In this mutant, h1 hybrids were not detected, indicating that they originated from incorporation of recycled glucosamine (*Figure 4—figure supplement 1*). Hybrids h2 and h3 were present in the peptidoglycan of the Δ*ampG* mutant as expected from the metabolic chart presented in *Figure 1—figure supplement 1*. Indeed, recycling of the tripeptide (hybrid h2) involves both the transport of anhydroMurNAc-peptides by AmpG and the transport of the tripeptide by the Opp permease (*Johnson et al., 2013*). Deletion of the *ampG* gene is therefore expected to reduce rather than abolish formation of the h2 hybrid. The h3 hybrid was proposed to originate from existing UDP-MurNAc-pentapeptide and de novo-synthesized GlcNAc moieties (above). Accordingly, the *ampG* deletion did not prevent formation of the h3 hybrid.

Analysis of the disaccharide-tetrapeptide (*Figure 4B* and *Supplementary file 2*) revealed a pattern of incorporation of unlabeled moieties very similar to that described above for the disaccharide-tripeptide. After the medium switch, the bulk of the $^{12}$C and $^{14}$N nuclei was incorporated into uniformly unlabeled disaccharide-tetrapeptides. Hybrid molecules retained labeled glucosamine moieties in GlcNAc or MurNAc (h1), a labeled tripeptide moiety in the tetrapeptide stem (h2), or labeling of the entire molecule except for GlcNAc (h3). As expected, the h1 hybrid was not detected in the Δ*ampG* mutant (*Figure 4—figure supplement 1*). The hybrid tetrapeptide stem present in h2 is in full agreement with the *E. coli* recycling pathway, as the tripeptide and the terminal D-Ala are expected to have different origins. Indeed, the terminal D-Ala of recycled tetrapeptide stems is cleaved off by a cytoplasmic L,D-carboxypeptidase to form a tripeptide that is added to MurNAc by the Mpl synthase

(*Templin et al., 1999*). In the following step, MurF adds the D-Ala-D-Ala dipeptide to generate the complete pentapeptide stem.

The isotopologues described above generated the isotopic clusters expected from their isotopic composition. The basis for the mirror image of the isotopic clusters of muropeptides exclusively generated from either labeled or unlabeled glucose and $NH_4Cl$ has already been introduced in the text and in *Figure 2*. The major h2 isotopologue mass peak had minor mass peaks both at lower and higher *m/z* values generating a seemingly symmetrical cluster (*Figure 4A and B*; *Figure 4—figure supplement 2* for simulated spectra). Peaks at lower *m/z* values mostly originated from the presence of rare $^{12}C$ and $^{14}N$ nuclei in the labeled moiety of the molecule. Peaks at higher *m/z* values mostly originated from the presence of rare $^{13}C$ and $^{15}N$ nuclei in the unlabeled moiety of the molecule. The contribution of these two effects shaped a nearly symmetrical isotopic cluster centered on the major isotopologue mass peak exclusively containing (i) $^{12}C$ and $^{14}N$ nuclei in the unlabeled moiety of the molecule, (ii) $^{13}C$ and $^{15}N$ nuclei in the labeled moiety, and (iii) $^{1}H$ and $^{16}O$ nuclei in the entire molecule.

## Kinetics of incorporation of unlabeled nuclei into muropeptide monomers

Our next objective was to compare the evolution of the ratios of uniformly labeled and unlabeled monomers following the culture medium switch. This analysis was based on the mass spectra of muropeptides isolated from culture samples withdrawn at various times after the medium switch as described in *Figure 3*. The mass spectra obtained for the disaccharide-tripeptide and disaccharide-tetrapeptide are displayed in *Figure 5A and C*, respectively. Variations in the relative abundance of the uniformly labeled and unlabeled isotopologues deduced from mass spectral ion intensities are presented in *Figure 5B and D*. For the disaccharide-tripeptide, replacement of the uniformly labeled isotopologue by its unlabeled counterpart reached ca. 50% in 85 min (*Figure 5B*). In contrast, the replacement was more rapid for the disaccharide-tetrapeptide (ca. 94% in 85 min) (*Figure 5D*). Thus, the isotopic labeling method resolved different kinetics of accumulation of the newly synthesized monomers (*Figure 5*) although their relative proportions in the peptidoglycan remained similar (*Figure 3*). The tetrapeptide and tripeptide stems may originate from sequential hydrolytic reactions involving the removal of D-Ala$^5$ from free (uncross-linked) pentapeptide stems by DDCs, followed by the removal of D-Ala$^4$ from the resulting tetrapeptide by the L,D-carboxypeptidase activity of YcbB (*Magnet et al., 2008*). The sequential nature of these reactions may explain, at least in part, the delayed accumulation of the disaccharide-tripeptide, as it derives from the disaccharide-tetrapeptide.

Kinetics of accumulation of hybrids revealed that hybrids h1 and h2, which originated from peptidoglycan recycling, were virtually absent at 5 min and increased between 5 and 85 min for the disaccharide-tripeptide and between 5 and 30 min for the disaccharide-tetrapeptide (*Figure 5*). This is expected since recycled building blocks (glucosamine and tripeptide for h1 and h2, respectively) originated from a peptidoglycan that was fully labeled at the time of the medium switch and remained partially labeled during the time course of the experiment. In contrast, h3 hybrids were already present at 5 min, and reached a maximum at ca. 18 and 30 min for tetrapeptide and tripeptide, respectively, as expected for their synthesis from a cytoplasmic pool of labeled UDP-MurNAc-pentapeptide present at the time of the medium switch.

## Impact of the medium switch on the isotopic composition of dimers

Three main types of isotopologues were observed after the medium switch for each of the four dimers (*Figure 6* for the Tri→Tri dimer and *Figure 6—figure supplements 1–3* for the Tetra→Tri, Tri→Tetra, Tetra→Tetra dimer, respectively). The first type of isotopologues were uniformly labeled. These dimers were generated by the cross-linking of stem peptides assembled before the medium switch. Thus, they correspond to fragments of the existing peptidoglycan. These dimers are referred to as old→old isotopologue dimers. The second type of isotopologues was generated by the cross-linking of a donor stem assembled after the medium switch to an acceptor stem assembled prior to the medium switch (referred as new→old isotopologue dimers). These isotopologue dimers correspond to a de novo synthesized donor stem attached to the existing peptidoglycan, which provided the acceptor. The third type of dimers was generated by the cross-linking of two neo-synthesized stem peptides (new→new). The h1, h2, and h3 hybrids depicted in *Figure 4* were assigned to the newly synthesized type of disaccharide-peptides since they contained unlabeled moieties implying

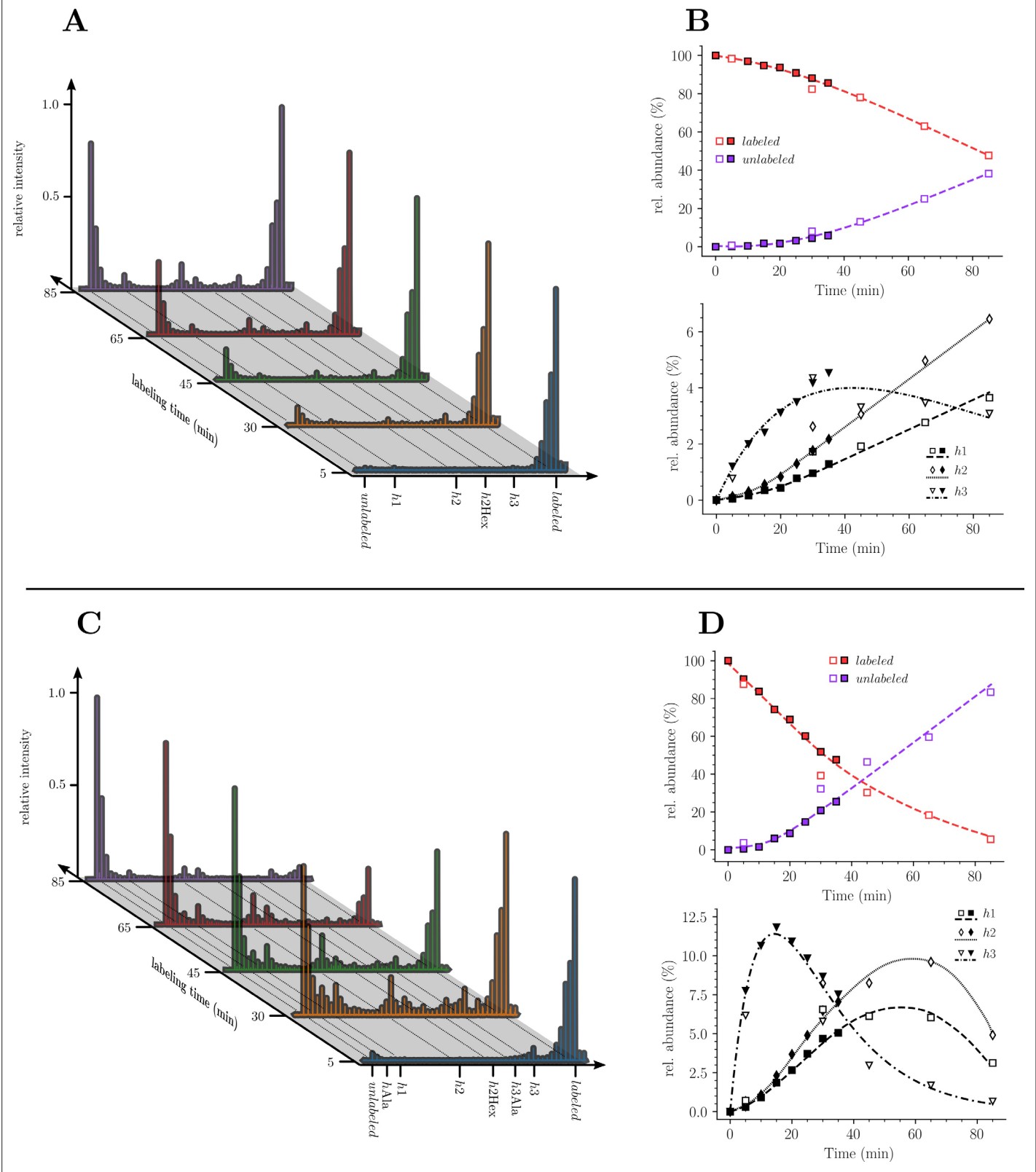

**Figure 5.** Kinetics analysis of the proportion of isotopologue monomers in the peptidoglycan of strain M1.5. (**A**) and (**C**), Mass spectra of disaccharide-tripeptide and disaccharide-tetrapeptide from bacteria collected at 5, 30, 45, 65, 85 min after the medium switch. (**B**) and (**D**), Kinetics analysis of the ratios of isotopologues calculated from relative ion current intensities for the disaccharide-tripeptide and disaccharide-tetrapeptide, respectively. The upper panels show the evolution of the relative abundance of uniformly labeled and unlabeled monomers. The lower panels present the evolution of

*Figure 5 continued on next page*

Figure 5 continued

the relative abundance of hybrids relative to the complete set of monomers. The open and solid symbols are the results from two biological repeats. See Supplementary data for the complete assignment of hybrid structures. The curves are interpolations (*Virtanen et al., 2020*).

their assembly after the medium switch. The absence of old→new dimers (see *Supplementary file 2* for MS/MS analyses) indicated that the peptide stems present in the existing peptidoglycan were not used as donors for cross-linking to neo-synthesized acceptors. For 4→3 cross-linked dimers, this result is expected since D,D-transpeptidases belonging to the PBP family use pentapeptide-containing donors, which are almost completely absent from mature peptidoglycan due to the hydrolysis of the D-Ala$^4$-D-Ala$^5$ amide bond by DDCs (See Introduction section and *Figure 1*). Accordingly, subunits containing pentapeptide stems were not detected (*Figure 3* and *Supplementary file 1a*). For the 3→3 cross-linked Tri→Tetra dimer, tetrapeptide stems present in the existing peptidoglycan could potentially be used as donors by the YcbB LDT although this was not observed. Thus, the absence of old→new 3→3 cross-linked Tri→Tetra dimer indicates that YcbB discriminates neo-synthesized tetrapeptide stems (used as donors) from existing tetrapeptide stems (used as acceptors) by an unknown mechanism that does not depend upon the structure of the disaccharide-peptide unit but upon the presence of the donor in a neo-synthesized glycan chain and of the acceptor in the existing peptidoglycan. The absence of old→new 3→3 cross-linked Tri→Tri dimers could be accounted for by the same mechanism.

Kinetic analyses revealed that the Tri→Tri and Tetra→Tri dimers formed by YcbB and PBPs, respectively, were almost exclusively of the new→old type at early times after the medium switch (i.e. ≤20 min) (*Figure 7A and B*). These dimers contained a tripeptide stem at the acceptor position. In contrast, Tri→Tetra and Tetra→Tetra (formed by YcbB and PBPs; *Figure 7C and D*, respectively), which contained a tetrapeptide stem at the acceptor position, were mixtures of new→new and new→old types of dimers at early times after the medium switch. Incubation for longer time periods resulted in an increase in the proportion of new→new isotopologues for all types of dimers. This increase was expected since serial and independent insertions of glycan strands at the same location result in the formation of new→new dimers. Note that this increase was more rapid for dimers containing a tetrapeptide stem at the acceptor position, as observed for tetrapeptide-containing monomers. As described above, this difference reflects the delayed formation of tripeptide stems from tetrapeptide stems by cleavage of D-Ala$^4$ by the L,D-carboxypeptidase activity of YcbB. At early times, tripeptide-containing new stems may therefore be present in small amounts.

Together, the results shown in *Figure 7* revealed that the kinetics of formation of isotopologues containing a tripeptide stem at the acceptor position were similar for the Tri→Tri and Tetra→Tri dimers generated by YcbB and PBPs, respectively. Likewise, the kinetics of formation of dimers containing a tetrapeptide stem at the acceptor position were similar for the Tri→Tetra and Tetra→Tetra generated by YcbB and PBPs, respectively. In contrast, distinct kinetics were observed between dimers containing a stem tripeptide or tetrapeptide in the acceptor position irrespective of the transpeptidase responsible for their formation (mainly new→old versus a mixture of new→new and new→old). Thus, the mode of insertion of peptidoglycan subunits depended upon the structure of the acceptor stem (tripeptide versus tetrapeptide) rather than upon the type of transpeptidase (PBPs versus YcbB). In addition, it is worth noting that the newly formed peptidoglycan subunits are more rapidly incorporated in dimers containing a tetrapeptide stem in the acceptor position than in dimers containing a tripeptide stem at this position.

In this study, we analyzed the participation of YcbB to global peptidoglycan synthesis following overproduction of this LDT and of the alarmone (p)ppGpp in combination with the inhibition of PBPs in certain experiments. Previous work concluded that YcbB also participates in the repair of peptidoglycan in a wild-type background (*Morè et al., 2019*). In addition to YcbB, this process required glycosyltransferase and DDC activities provided by PBP1b and PBP6a, respectively. This implies that peptidoglycan repair proceeds by polymerization of new glycan chains that are cross-linked by PBPs. It is worth noting that the mode of insertion of new peptidoglycan subunits delineated in the current study is fully compatible with this model since we detected only one type of 3→3 cross-linked dimers (new→old and not old→new).

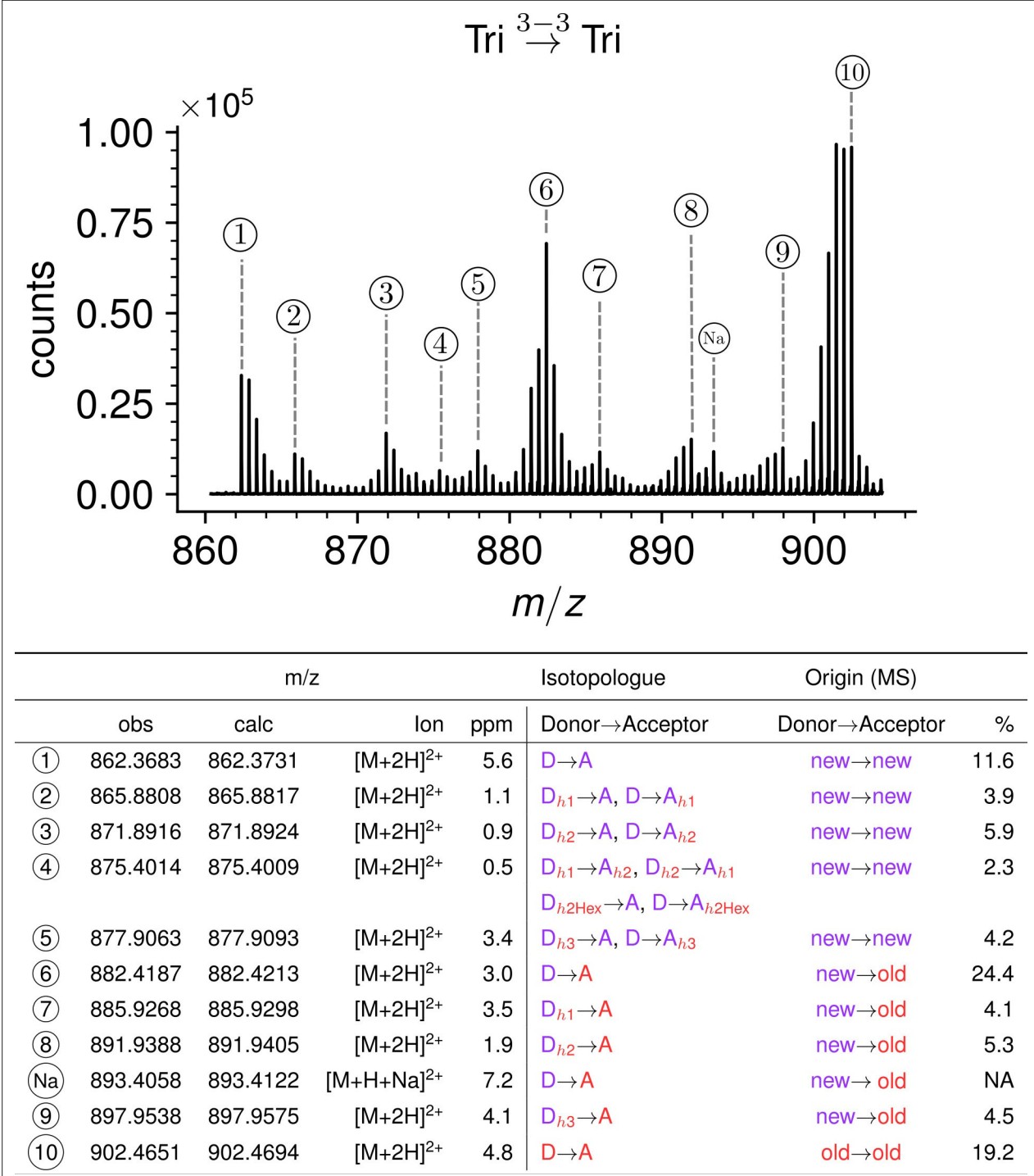

| | m/z | | | | Isotopologue | Origin (MS) | |
|---|---|---|---|---|---|---|---|
| | obs | calc | Ion | ppm | Donor→Acceptor | Donor→Acceptor | % |
| ① | 862.3683 | 862.3731 | [M+2H]²⁺ | 5.6 | D→A | new→new | 11.6 |
| ② | 865.8808 | 865.8817 | [M+2H]²⁺ | 1.1 | $D_{h1}$→A, D→$A_{h1}$ | new→new | 3.9 |
| ③ | 871.8916 | 871.8924 | [M+2H]²⁺ | 0.9 | $D_{h2}$→A, D→$A_{h2}$ | new→new | 5.9 |
| ④ | 875.4014 | 875.4009 | [M+2H]²⁺ | 0.5 | $D_{h1}$→$A_{h2}$, $D_{h2}$→$A_{h1}$ | new→new | 2.3 |
| | | | | | $D_{h2Hex}$→A, D→$A_{h2Hex}$ | | |
| ⑤ | 877.9063 | 877.9093 | [M+2H]²⁺ | 3.4 | $D_{h3}$→A, D→$A_{h3}$ | new→new | 4.2 |
| ⑥ | 882.4187 | 882.4213 | [M+2H]²⁺ | 3.0 | D→A | new→old | 24.4 |
| ⑦ | 885.9268 | 885.9298 | [M+2H]²⁺ | 3.5 | $D_{h1}$→A | new→old | 4.1 |
| ⑧ | 891.9388 | 891.9405 | [M+2H]²⁺ | 1.9 | $D_{h2}$→A | new→old | 5.3 |
| Na | 893.4058 | 893.4122 | [M+H+Na]²⁺ | 7.2 | D→A | new→ old | NA |
| ⑨ | 897.9538 | 897.9575 | [M+2H]²⁺ | 4.1 | $D_{h3}$→A | new→old | 4.5 |
| ⑩ | 902.4651 | 902.4694 | [M+2H]²⁺ | 4.8 | D→A | old→old | 19.2 |

**Figure 6.** Structure of Tri→Tri isotopologues in the peptidoglycan of strain M1.5 grown in the absence of β-lactams. Mass spectrum highlighting the 10 major isotopologues. Structure of the 10 major isotopologues. The observed (obs) and calculated (cal) *m/z* values of the ions of interest (Ion) and the difference in part per million (ppm) between these values are indicated. New (neo-synthesized) unlabeled moieties of isotopologues are indicated in purple. Old (existing) labeled moieties of isotopologues are indicated in red. h1, h2, and h3 (in red) refer to recycled moieties originating from existing labeled peptidoglycan (**h1 and h2**) or from the existing (labeled) UDP-MurNAc-pentapeptide pool as described in *Figure 4*. The origin of the donor and acceptor participating in the cross-linking reaction, new (neo-synthesized) or old (existing in the cell wall), are indicated in purple and red, respectively. The relative abundance of the isotopologues (%) was deduced from the relative intensity of peaks corresponding to [M+2 H]²⁺ ions as labeled in the upper panel. Abbreviations: h1, hybrid containing a recycled glucosamine moiety; h2, hybrid containing a recycled tripeptide moiety;

*Figure 6 continued on next page*

*Figure 6 continued*

h3, hybrid containing a labeled MurNAc-tripeptide moiety originating from the existing UDP-MurNAc-pentapeptide pool. NA, not applicable, as the sodium adduct was not taken into consideration.

The online version of this article includes the following figure supplement(s) for figure 6:

**Figure supplement 1.** Structure of Tetra→Tri isotopologues in the peptidoglycan of strain M1.5 grown in the absence of β-lactams.

**Figure supplement 2.** Structure of Tri→Tetra isotopologues in the peptidoglycan of strain M1.5 grown in the absence of β-lactams.

**Figure supplement 3.** Structure of Tetra→Tetra isotopologues in the peptidoglycan of strain M1.5 grown in the absence of β-lactams.

## Mode of insertion of peptidoglycan subunits in the absence of any LDT

Kinetic analyses of the insertion of peptidoglycan subunits was analyzed in *E. coli* Δ6*ldt*, a derivative of strain BW25113 that did not harbor any of the six *E. coli* LDT genes. The absence of the corresponding enzymes greatly simplifies both the metabolic scheme for peptidoglycan cross-linking and the muropeptide *rp*HPLC profile (*Figure 8*, *Figure 1—figure supplement 3*). This profile contains only two main peaks, corresponding to the tetrapeptide monomer and the Tetra→Tetra dimer due to the absence of the formation of dimers containing 3→3 cross-links (LDT activity) and of monomers

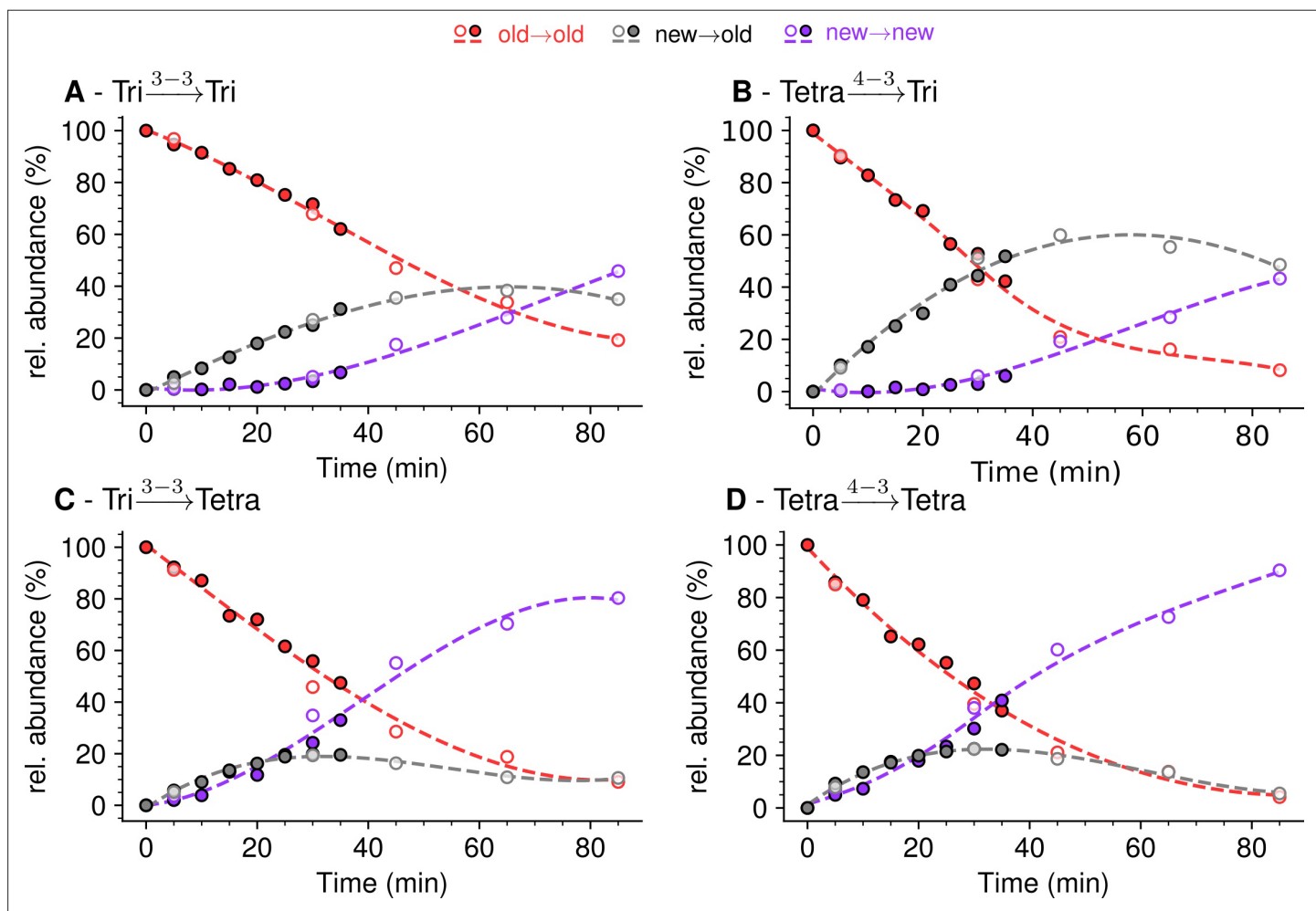

**Figure 7.** Mode of cross-linking of de novo synthesized peptidoglycan subunits in *Escherichia coli* M1.5 grown in the absence of β-lactam. For each dimer (**A**), (**B**), (**C**), and (**D**), the relative abundance (rel. abundance) was calculated for isotopologues originating from cross-linking of (**i**) two existing stems (old→old; red), (ii) a new donor stem (i.e. a stem synthesized after the medium switch) and an existing acceptor stem (new→old; gray), and (iii) two new stems (new→new; purple). The calculations take into account the contribution of de novo synthesized stems originating from recycled moieties and from the labeled UDP-MurNAc-pentapeptide pool as described in *Figure 6* and in *Figure 6—figure supplements 1–3* for the Tri→Tri, Tetra→Tri, Tri→Tetra, and Tetra→Tetra dimers, respectively. The open and closed symbols correspond to data from two independent experiments. The curves are interpolations (*Virtanen et al., 2020*).

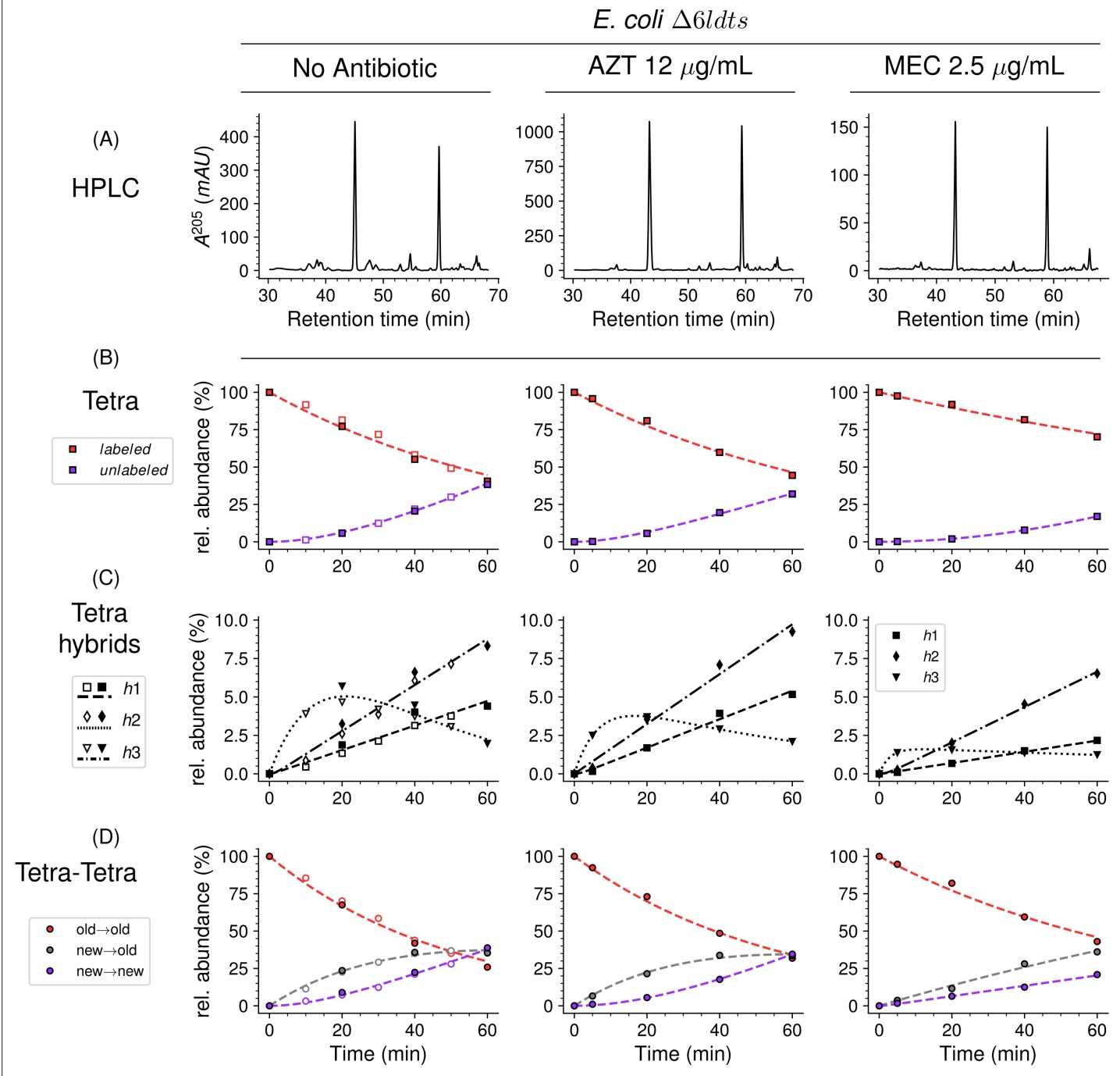

**Figure 8.** Mode of insertion of peptidoglycan subunits in the absence of L,D-transpeptidases and impact of the inhibition of penicillin-binding proteins (PBPs) by aztreonam and mecillinam. (**A**) Muropeptide *rp*HPLC profile of PG from *Escherichia coli* strain Δ6*ldt* grown in the absence of β-lactam or in the presence of aztreonam (AZT) or mecillinam (MEC). Kinetic data were collected for fully labeled (red), fully unlabeled (purple) tetrapeptide monomers (Tetra) (**B**) and hybrids (**C**) defined in *Figure 6—figure supplement 3*. For the Tetra→Tetra dimer (**D**), the relative abundance (rel. abundance) was calculated for isotopologues originating from cross-linking of (**i**) two existing stems (old→old; red), (ii) a new donor stem (i.e. a stem synthesized after the medium switch) and an existing acceptor stem (new→old; gray), and (iii) two new stems (new→new; purple). The calculations take into account the contribution of de novo synthesized stems originating from recycled moieties and from the labeled UDP-MurNAc-pentapeptide pool. The open and closed symbols correspond to data from two independent experiments. The curves are interpolations (*Virtanen et al., 2020*).

The online version of this article includes the following figure supplement(s) for figure 8:

**Figure supplement 1.** 100× phase contrast images of strain BW25113 grown for 60 min in the absence of β-lactams (**A**) or in the presence of aztreonam at 12 µg/mL (**B**) or mecillinam at 2.5 µg/mL (**C**).

*Figure 8 continued on next page*

*Figure 8 continued*

**Figure supplement 2.** Comparison of the mode of insertion of peptidoglycan subunits in *Escherichia coli* strain Δ6*ldt* that does not produce any L,D-transpeptidase (**A**), (**B**), and (**C**) and in the parental 'wild-type' strain BW25113 (**D**), (**E**), and (**F**).

**Figure supplement 3.** Acceptor-to-donor ratio of neo-synthesized stems (ADRNS) of 'wild-type' *Escherichia coli* BW25113 and strain Δ6ldt.

**Figure supplement 4.** Kinetic analysis of the isotopologue content of monomers from *Escherichia coli* BW25113 (left) and M1.5 (right).

or dimers containing tripeptide stems (L,D-carboxypeptidase activity). The medium switch led to the accumulation of the unlabeled tetrapeptide monomer and of hybrids h1, h2, and h3 (*Figure 8*). The majority of the Tetra→Tetra dimers of the Δ6*ldt* strain were hybrids containing an unlabeled (de novo synthesized) tetrapeptide stem at the donor position and a labeled tetrapeptide stem (existing peptidoglycan) at the acceptor position. This pattern mainly fits the one-at-a time mode of insertion of newly synthesized glycan strands (*Figure 1—figure supplement 2*).

## Mode of insertion of peptidoglycan subunits into the septum and in the lateral wall

In *E. coli*, two β-lactams, mecillinam and aztreonam, specifically inhibit the synthesis of the lateral cell wall and of the septum by inactivating the D,D-transpeptidase activity of PBP2 and PBP3, respectively. Aztreonam (12 µg/mL) and mecillinam (2.5 µg/mL) were added to the labeled culture medium of strain Δ6*ldt* 5 min prior to the medium switch and to the unlabeled culture medium used to resuspend the bacteria at the medium switch. Aztreonam and mecillinam were used at inhibitory concentrations corresponding to 200- and 20-fold the minimal inhibitory concentration of the drugs, respectively (*Supplementary file 1c*). Exposure to aztreonam or mecillinam resulted in the morphological changes expected for the inhibition of septum formation (filamentation) or side wall synthesis (rounding), respectively (*Figure 8—figure supplement 1*). The kinetics of insertion of de novo synthesized subunits in the tetrapeptide monomers and in the Tetra→Tetra dimers were similar in the cultures containing aztreonam or no drug (*Figure 8*). The pattern corresponded to the one-at-a time mode of insertion of newly synthesized glycan strands since the new→new dimers were not abundant at the beginning of the kinetics. In contrast, the new→new and new→old isotopologue dimers were both present in the cultures containing mecillinam, indicating that formation of the septum involves insertion of multiple strands at a time.

## Mode of insertion of peptidoglycan subunits in conditions in which YcbB is the only functional transpeptidase

In the presence of ampicillin (16 µg/mL), the peptidoglycan of strain M1.5 was predominantly (ca. 99%) cross-linked by YcbB since the D,D-transpeptidase activity of PBPs was inhibited by the drug (*Figure 9* and *Supplementary file 1a*). Inhibition of DDCs belonging to the PBP family prevented full hydrolysis of D-Ala[5] since pentapeptide stems were detected both in monomers and in the acceptor position of dimers. New→new rather than new→old isotopologues were predominantly detected among dimers containing a tetrapeptide or a pentapeptide stem in the acceptor position. These dimers originate from neo-synthesized glycan chains cross-linked to each other. In contrast, the new→old isotopologues were more abundant than the new→new isotopologues in the Tri→Tetra dimers from the peptidoglycan of M1.5 grown in the absence of ampicillin at early times (*Figure 7*). This difference was not observed for the Tri→Tri dimers since the new→old isotopologues were prevalent for growth of M1.5 both in the presence or absence of ampicillin.

## Mode of insertion of peptidoglycan subunits in a bacterial strain harboring the wild-type complement of PBP and LDT genes

Our last objective was to characterize the mode of peptidoglycan synthesis in a wild-type background with respect to genes involved in peptidoglycan synthesis. Toward this aim, we analyzed the peptidoglycan from the parental 'wild-type' strain BW25113. *Figure 8—figure supplement 2* provides a comparison of the kinetics obtained for both the BW25113Δ6*ldt* and parental BW25113 strains grown in the absence of β-lactams and in the presence of aztreonam or mecillinam. This analysis showed that the conclusions drawn from the analyses performed with strain BW25113Δ6*ldt* (*Figure 8*) are not biased by the deletion of the six *ldt* genes. The acceptor-to-donor ratio of neo-synthesized stems

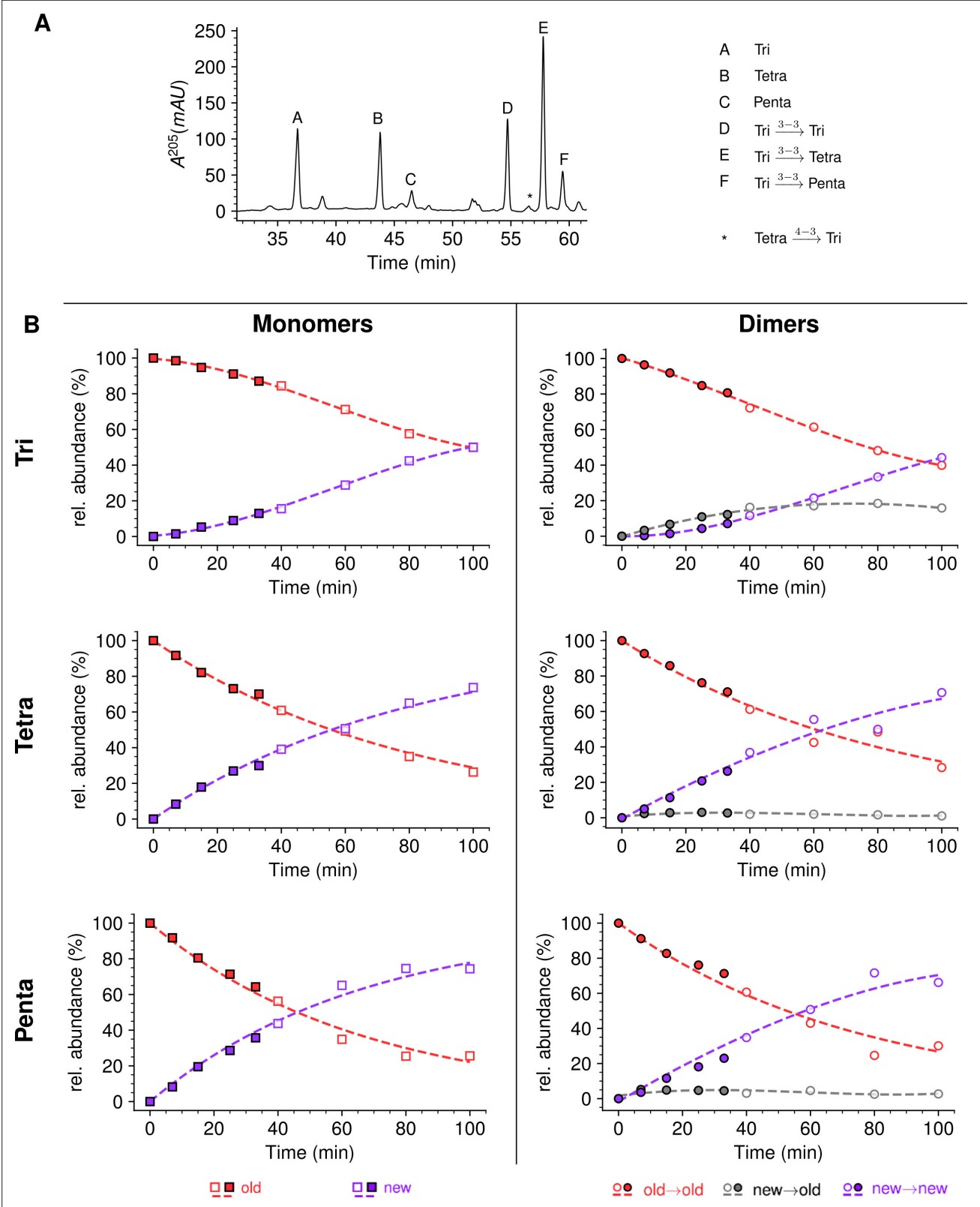

**Figure 9.** Mode of insertion of peptidoglycan in conditions in which YcbB is the only functional transpeptidase following inhibition of the D,D-transpeptidase activity of penicillin-binding proteins (PBPs) by ampicillin. (**A**) Muropeptide *rp*HPLC profile of PG from *Escherichia coli* strain M1.5 grown in the presence of ampicillin (16 µg/mL), which was added 5 min prior to the medium switch and to the unlabeled culture medium used for the medium switch corresponding to *t*=0. (**B**) Kinetic data were collected for newly synthesized (new, purple) and existing (old, red) monomers.Kinetic data were also

*Figure 9 continued on next page*

*Figure 9 continued*

collected for dimers originating from cross-linking of (**i**) two existing stems (old→old; red), (ii) a new donor stem (i.e. a stem synthesized after the medium switch) and an existing acceptor stem (new→old; gray), and (iii) two new stems (new→new; purple). The calculations take into account the contribution of de novo synthesized stems originating from recycled moieties and from the labeled UDP-MurNAc-pentapeptide pool. The open and closed symbols correspond to data from two independent experiments. Rel. abundance, relative abundance of isotopologues. Disaccharide-peptide subunits contained a tripeptide (Tri), a tetrapeptide (Tetra), or a pentapeptide (Penta) stem. The curves are interpolations (*Virtanen et al., 2020*).

(ADRNS) values were similar for both strains (*Figure 8—figure supplement 3*) indicating that the mode of insertion of new stems in the Tetra→Tetra dimers is not affected by the deletion of the six *ldt* genes. For the sake of completeness, we also compared the kinetics of appearance of muropeptides resulting from the activity of LDTs that are present in low amount in the parental strain BW25113 and are more abundant in mutant M1.5 due to overproduction of YcbB (*Supplementary file 1a*). Qualitatively, the kinetics were similar for muropeptides extracted from these two strains (*Figure 8—figure supplement 4*). Overall, the replacement of old by new muropeptides was more rapid for the 'wild-type' strain than for mutant M1.5.

## Discussion

Previous studies of the expansion of the peptidoglycan macromolecule relied on labeling with [³H] or [¹⁴C]DAP (*de Jonge et al., 1989*). Quantitatively, the number of glycan strands inserted at a time was estimated by determining the acceptor-to-donor radioactivity ratio (ADRR), that is, the radioactivity ratio in the acceptor and donor positions of dimers (*Figure 1—figure supplement 2*; *Burman and Park, 1984*). By analogy, we define here the acceptor-to-donor ratio in neo-synthesized stems (ADRNS). ADRR and ADRNS determinations rely on opposite labeling schemes involving pulse labeling with radioactive DAP versus incorporation of newly synthesized (unlabeled) subunits into the fully labeled peptidoglycan, respectively. The latter strategy was chosen because of the ease of obtaining fully labeled cells with non-radioactive heavy isotopes. Switching the medium from labeled to unlabeled was chosen, rather than the unlabeled-to-labeled switch, since this facilitates the detection of neo-synthesized isotopologues present at a low abundance at early times after the medium switch. This is due to the fact that muropeptides from the existing peptidoglycan, which are preponderant at the beginning of the kinetics, are fully labeled and hence have higher *m/z* values than those of muropeptides containing ¹²C and ¹⁴N light isotopes following the medium switch. Consequently, sodium adducts to the fully labeled muropeptides do not appear in the relevant region of the chromatogram that contains muropeptides of lower *m/z* value due to the incorporation of light isotopes.

Pulse labeling with radioactive DAP suffers from four major limitations that are not encountered in our labeling strategy. First, optimization of radioactive DAP incorporation requires mutations that prevent endogenous DAP synthesis

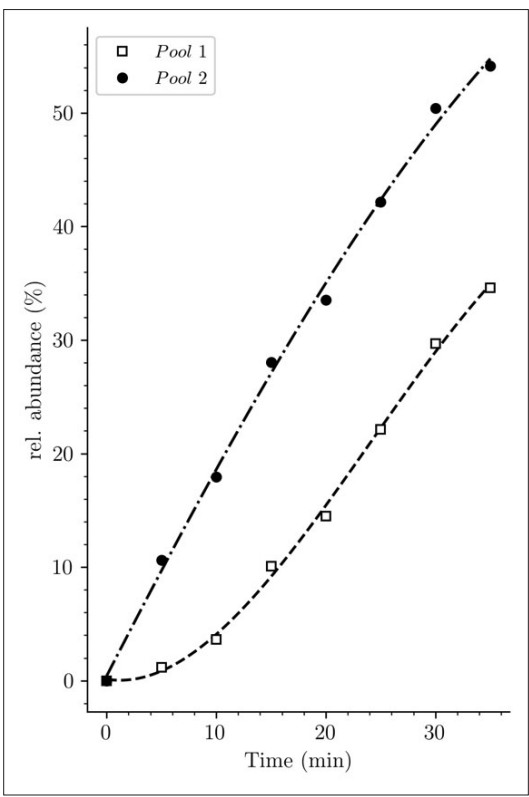

**Figure 10.** Potential origin of the lag phase previously observed in experiments based on incorporation of radioactive diaminopimelic acid (DAP). The accumulation kinetics of neo-synthesized disaccharide-tetrapeptide isotopologues containing unlabeled DAP (pool 1, squares) was compared to that of all neo-synthesized disaccharide-tetrapeptide isotopologues, including hybrids h2 and h3 (pool 2, solid circles) (data from strain Δ6*ldt* grown in the absence of β-lactam). The curves are interpolations (*Virtanen et al., 2020*).

(*lysA*) and conversion of DAP to L-Lys (*dapF*) (*Burman and Park, 1984*; *Glauner and Höltje, 1990*; *Goodell, 1985a*; *Goodell and Schwarz, 1985b*; *Figure 1—figure supplement 1*). Starvation prior to the addition of radiolabeled DAP was used to minimize the lag between the addition of radioactive DAP into the culture medium and its incorporation into peptidoglycan (*Glauner and Höltje, 1990*; *Burman and Park, 1984*). However, depleting the intracellular DAP pool compromised the integrity of the peptidoglycan layer (*Burman et al., 1983b*), affected cell morphology (*Verwer and Nanninga, 1980*), and reduced turnover (*Burman et al., 1983b*). Our labeling scheme is not affected by these confounding factors since it does not require any metabolic engineering. A second limitation of the DAP labeling procedures is their inability to discriminate between muropeptides originating from the existing peptidoglycan, from the recycled L-Ala-D-iGlu-DAP tripeptide, or from the existing UDP-MurNAc-pentapeptide pool, because all of them contain non-radioactive DAP. In contrast, our procedure easily discriminates between these three types of muropeptides since they correspond to different isotopologues, namely uniformly labeled subunits and hybrids h2 and h3, respectively (*Figure 4*). The latter hybrids are likely to contribute to the lag phase observed for the incorporation of radioactive DAP into peptidoglycan in previous reports (*Glauner and Höltje, 1990*; *Burman and Park, 1984*). Accordingly, a lag phase was observed in our study when h2

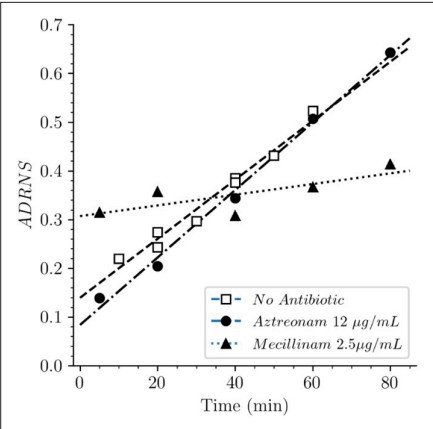

**Figure 11.** Acceptor-to-donor ratio of neo-synthesized stems (ADRNS) of *Escherichia coli* Δ6*ldt*. Bacteria were grown in the presence of aztreonam, mecillinam, or in the absence of drug. Curves are obtained by linear regression. ADRNS were deduced from the relative abundance of new→old and new→new dimers appearing in *Figure 8*. Please note that the ADRNS was the highest for growth in the presence of mecillinam (this figure) in spite of the fact that the overall rate of replacement of old stems by new stems was the lowest for this growth condition, as evidenced by the slow rise of both the new→old and new→new dimers (*Figure 8*). This apparent discrepancy is accounted for by the fact that the ADRNS is a ratio solely based on neo-synthesized dimers.

and h3 hybrids containing [$^{13}$C]- and [$^{15}$N]-labeled DAP were excluded from the dataset (*Figure 10*). A third limitation of the radioactive DAP labeling procedures for ADRR determination originates from the fact that they provide access only to the average radioactivity content of donor and acceptor stem peptides. Indeed, these procedures are based on purification of individual dimers, derivatization of their free amino group located in the acceptor stem, acid hydrolysis, and chromatography to separate DAP and derivatized DAP originating from the donor and acceptor stems, respectively (*Burman et al., 1983a*). Our procedure is richer in information because it provides direct access to the structure of dimers enabling identification of new→new, new→old, old→new, and old→old dimers based on differences in mass and fragmentation patterns. The fourth limitation stems from the fact that exploring peptidoglycan recycling by the radioactive DAP labeling procedures requires use of specific mutants and study designs. For example, *Jacobs et al., 1994*, used mutants deficient in the production of the AmpG and Opp permeases in which peptidoglycan fragments are not recycled in the cytoplasm but released in the culture medium (*Jacobs et al., 1994*). In contrast, our approach enables investigating peptidoglycan recycling and the mode of peptidoglycan expansion in a single experiment in the absence of any metabolic perturbation since it does not require altering the recycling pathway and is solely based on the determination of the structure of peptidoglycan fragments.

Strain BW25113Δ6*ldt* offered the possibility to investigate the mode of insertion of new subunits by PBPs in the absence of any LDT. Aztreonam and mecillinam were used to specifically inhibit synthesis of septal and lateral peptidoglycan, respectively. In the presence of aztreonam, the ADRNS for side wall synthesis was 0.14, 5 min after the medium switch and increased to 0.20, 0.34, 0.50, and 0.64 at 20, 40, 60, 80 min, respectively (*Figure 11*). At this time scale, existing disaccharide-peptide units are gradually replaced by newly synthesized disaccharide-peptides accounting for the increase in new→new dimers and the increase in the ADRNS value. The extrapolation of the ADRNS at *t*=0 (time of the medium switch) is therefore relevant to the evaluation of the mode of insertion of new subunits

into the peptidoglycan. The extrapolated ADRNS value of 0.08 (*Figure 8—figure supplement 3*) was similar to the ADRR value of 0.10 reported by de Jonge and collaborators based on extrapolating data to a synchronized culture exclusively containing elongating cells (*de Jonge et al., 1989*). In the presence of mecillinam, the ADRNS values marginally increased over time, providing an extrapolated value of 0.31 for $t$=0. This value is lower than the ADRR of 0.64 obtained by extrapolating data to a synchronized culture exclusively containing constricting cells (*de Jonge et al., 1989*). The origin of this difference is unknown but could involve the fact that blocking cell elongation by mecillinam may not fully prevent incorporation of newly synthesized subunits into the peptidoglycan since this drug does not prevent the polymerization of glycan chains by the elongasome (*Uehara and Park, 2008*). Together, our results are in agreement with previous analyzes (*de Jonge et al., 1989*) that have shown single-strand insertion of newly synthesized peptidoglycan into the side wall versus insertion of multiple strands for septal peptidoglycan synthesis. In the absence of mecillinam and aztreonam, the ADRNS value (0.14) was intermediate between the values observed in the presence of aztreonam (0.08) and mecillinam (0.31), as qualitatively expected for the combined contributions of cross-links located in the side wall and in the septum and the lower contribution of the latter to the cell surface.

In the absence of β-lactam, *E. coli* M1.5 relied both on the D,D-transpeptidase activity of PBPs and the LDT activity of YcbB for peptidoglycan cross-linking, leading to a high peptidoglycan content in both 4→3 (56%) and 3→3 (44%) cross-linked dimers (*Supplementary file 1a*). The proportions of dimers containing two newly synthesized stems (new→new) or one newly synthesized and one existing stem (new→old) were similar for the two types of cross-links (*Figure 7*). Thus, the D,D-transpeptidase activity of PBPs and the LDT activity of YcbB were involved in similar modes of insertion of new strands into the existing peptidoglycan of strain M1.5. The ADRNS values were in the order of 0.17 and 0.22 for 4 → 3 and 3 → 3 cross-links, respectively. These values indicate that the neo-synthesized glycan strands were predominantly but not exclusively cross-linked to the existing peptidoglycan for both types of transpeptidation reactions. Peptidoglycan polymerization might involve single-strand insertion into the side wall and insertion of multiple strands into the septum, as discussed above for strain Δ6*ldt*. The striking similarities between dimers generated by PBPs and YcbB raise the possibility that the mode of insertion of new subunits is not determined by the transpeptidases but by other components of the peptidoglycan polymerization complexes, such as the scaffolding proteins.

The peptidoglycan of strain M1.5 grown in the presence of ampicillin was mainly (99%) cross-linked by YcbB following the inhibition of the D,D-transpeptidase activity of PBPs (*Figure 9*). Strikingly,

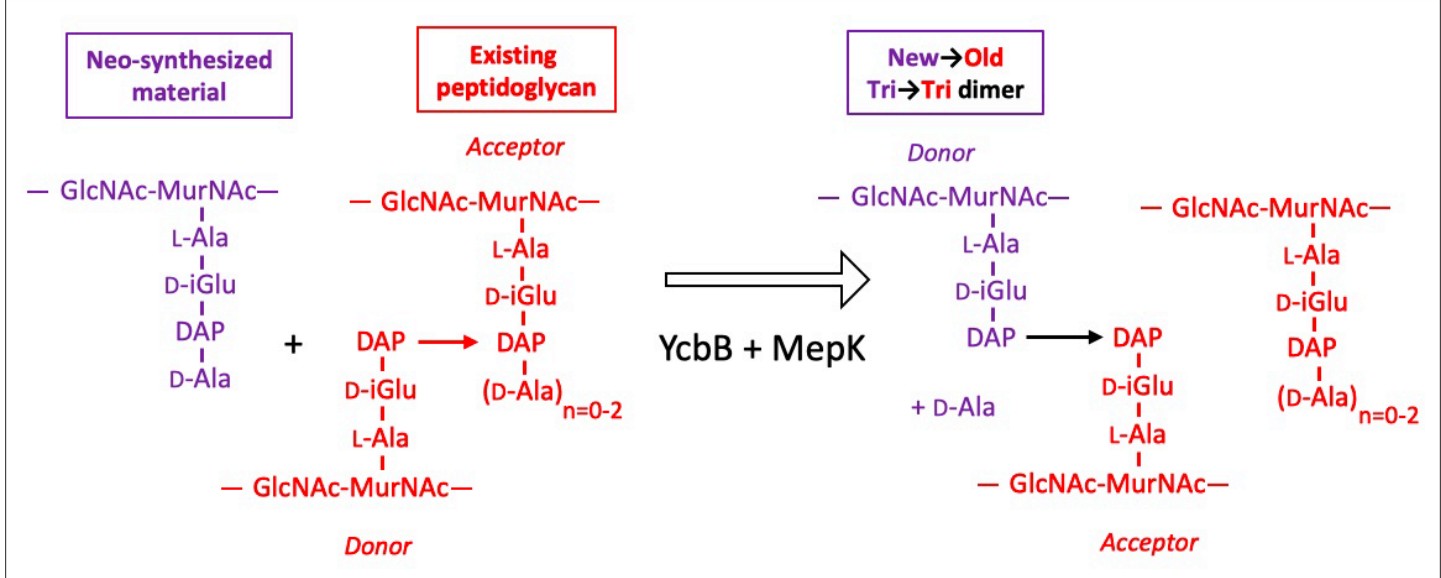

**Figure 12.** Model for the cross-linking of neo-synthesized material to existing peptidoglycan in strain M1.5 grown in the presence of ampicillin. The formation of new→old cross-links results from cleavage of existing 3→3 cross-links by endopeptidase MepK thereby providing the tripeptide acceptor of the cross-linking reaction catalyzed by YcbB. Please note that the stem tripeptide in the donor position of dimers in the existing peptidoglycan is used as an acceptor in the neo-synthesized cross-links. This accounts for the fact that new→old dimers are exclusively of the Tri→Tri type. The acceptor stem in the existing peptidoglycan may harbor 0, 1, or 2 D-Ala residue ($n$=0–2).

the Tri→Tetra and Tri→Penta dimers were almost exclusively of the new→new type. This pattern was not found in strains M1.5 and Δ6*ldt* grown in the absence of β-lactam (**Figures 7 and 8**). The prevalence of new→new dimers may be a consequence of the inhibition of the transpeptidase activity of PBPs by ampicillin that results in the uncoupling of the cross-linking and transglycosylation reactions (**Uehara and Park, 2008**; **Cho et al., 2014**). This in turn results in the accumulation of linear glycan strands that are eventually cross-linked to each other by YcbB because they are not effectively degraded due to a mutation present in M1.5 that prevents production of lytic transglycosylase Slt70 (**Voedts et al., 2021**; **Cho et al., 2014**). The resulting new→new cross-linked material was almost exclusively connected to the existing peptidoglycan by the formation of Tri→Tri dimers because new→old isotopologues were only detected in this type of dimers. This implies that the acceptors of the cross-linking reactions in the existing peptidoglycan were tripeptides, whereas existing tetrapeptides also acted as acceptors in the absence of ampicillin (**Figure 7**). The nearly exclusive use of existing tripeptides as acceptors in the presence of ampicillin could be accounted for by the fact that these tripeptides originate from cleavage of 3→3 cross-links liberating the tripeptide stems present in the donor position of the existing dimers. This model implies a cooperation between YcbB and an endopeptidase that provides the tripeptide acceptor substrate of the reaction by cleaving existing 3→3 cross-links (**Figure 12**). This enzyme could be MepK since this endopeptidase preferentially cleaves 3→3 cross-links in vitro, is essential for bacterial growth only in strains overproducing YcbB, and is not inhibited by ampicillin (**Voedts et al., 2021**). According to this model, the reactions catalyzed by MepK and YcbB are concerted rather than sequential. This mode of expansion of the peptidoglycan network differs from the 'make-before-break' models that postulate that neo-synthesized glycan strands are cross-linked to free stem peptides of the existing peptidoglycan prior to the cleavage of existing cross-links. In the absence of ampicillin, the hydrolysis of 4→3 cross-links by endopeptidases belonging to the PBP family could generate tetrapeptide acceptors in addition to the tripeptide acceptors exclusively used in the presence of the drug.

   **Cho et al., 2014** proposed that the antibacterial activity of β-lactams does not merely rely on a loss of function, that is, the inactivation of the D,D-transpeptidase activity of PBPs by acylation of their catalytic Ser residue, but by inducing a toxic malfunctioning of the peptidoglycan biosynthesis machinery. This involves uncoupling of transglycosylation and transpeptidation, accumulation of

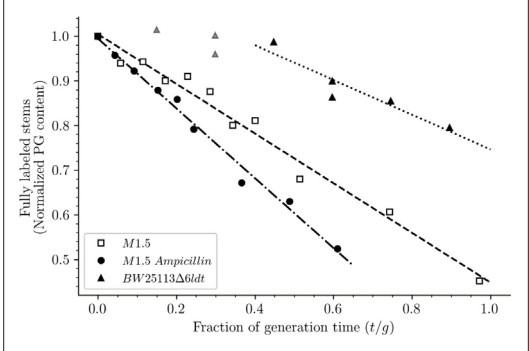

**Figure 13.** Release of peptidoglycan fragments from sacculi. Kinetics of the isotopologue composition of sacculi was used to determine the extent of the release of peptidoglycan fragments from sacculi. For this purpose, we considered that the decrease in the number of fully labeled disaccharide-peptide subunits is equal to the number of subunits released from the peptidoglycan. This is legitimate since peptidoglycan subunits are recycled in a large number of moieties including two glucosamine residues, two acetyl groups, one D-lactoyl residue, the L-Ala-D-iGlu-DAP tripeptide or its constitutive amino acids, and two D-Ala residues (**Figure 1—figure supplement 1**). Disaccharide-peptide units issued from recycling are therefore mixtures of labeled and unlabeled moieties that can be easily distinguished from fully labeled subunits. Of note, release of peptidoglycan fragments in the culture medium, corresponding to a small proportion of peptidoglycan degradation products (**Goodell and Schwarz, 1985b**), also results in a reduction in the content of fully labeled disaccharide-peptides. Thus, the decrease in the absolute number of fully labeled disaccharide-peptide subunits in one generation directly provides an estimate of the fraction of the existing peptidoglycan that is degraded during each generation. For this reason, the decrease in the relative amount of fully labeled peptidoglycan disaccharide-peptides (normalized to 1.0 at the medium switch) was computed as a function of time (normalized for the generation time; **Supplementary file 2**-table S2). The decrease was linear for M1.5 grown in the presence or absence of ampicillin. For BW25113Δ6*ldt*, a decrease in the fully labeled stems was only observed after 0.45 generation time. For this reason, regression analysis was performed for a portion of the kinetics (from 0.45 to 0.85 generation time; data points excluded from regression analysis are shown as gray triangles). The estimates of combined peptidoglycan turnover and recycling per generation were 39% for strain Δ6*ldt* and 56% or 78% for M1.5 grown in the absence or presence of ampicillin, respectively.

polymerized glycan chains, and their degradation by lytic transglycosylase Slt70 (*Banzhaf et al., 2012*; *Cho et al., 2014*; *Uehara and Park, 2008*). The resulting futile cycle depletes resources and prevents bacterial growth. Exposure of Δ*sltY* cells to mecillinam was reported to result in the polymerization of an 'aberrant peptidoglycan' enriched in 3→3 cross-links. In comparison to this previous report (*Cho et al., 2014*), our analysis revealed that overproduction of YcbB results in substantially different modifications of peptidoglycan metabolism. First, M1.5 grown in the absence of β-lactam released a high proportion of its peptidoglycan per generation (56%) (*Figure 13*). This indicates that formation of 3→3 cross-links by YcbB is sufficient to stimulate peptidoglycan degradation in the absence of β-lactams. Under these conditions, Slt70 was not the main lytic enzyme since this activity is impaired in M1.5 (*Voedts et al., 2021*). Exposure to β-lactam had an additional stimulating effect on peptido-glycan degradation that reached 78% per generation for M1.5 grown in the presence of ampicillin (*Figure 13*). Second, overproduction of YcbB enabled growth of M1.5 in the presence of β-lactams indicating that the 'aberrant peptidoglycan' described by *Cho et al., 2014*, can in fact sustain bacte-rial growth. This involved an unprecedented mode of peptidoglycan synthesis implicating polym-erization of linear glycan chains, their cross-linking by YcbB, and the incorporation of the resulting neo-synthesized material into the existing peptidoglycan by the concerted action of LDT YcbB and of β-lactam-insensitive endopeptidase MepK.

## Conclusions

In conclusion, we report an innovative method for exploring peptidoglycan metabolism that does not depend upon metabolic engineering for labeling and is therefore applicable to various genetic backgrounds. The choice of full labeling with $^{13}$C and $^{15}$N enabled us to investigate peptidoglycan metabolism at a very fine level of detail since most isotopologues predicted to occur according to known recycling and biosynthesis pathways were detected and kinetically characterized. In addition, high-resolution MS and MS/MS identified the donors and acceptors that participated in individual cross-linking reactions rather than the average composition of dimers. This revealed an unexpected diversity in the modes of insertion of glycan strands into the expanding peptidoglycan macromolecule in the presence and absence of β-lactams. In particular, our results suggest that the reactions catalyzed by YcbB and by 3→3 specific endopeptidases in the presence of β-lactams are concerted rather than sequential as proposed by the 'make-before-break' models of peptidoglycan expansion. As previ-ously discussed (*Cho et al., 2014*), efforts in drug discovery should focus on molecules that induce a lethal malfunctioning of multiple cellular targets rather simple inhibitors. This implies that candidate molecules should be tested for their mode of action. The mass data analysis pipeline described in this study should facilitate such screening efforts by providing specific signatures for molecules acting on peptidoglycan synthesis.

## Materials and methods
### Sample preparation and analyses
#### Strains

*E. coli* M1.5 (*Hugonnet et al., 2016*; *Voedts et al., 2021*) is a derivative of *E. coli* BW25113 (*Datsenko and Wanner, 2000*) harboring (i) plasmid pJEH12 for IPTG-inducible expression of the *ycbB* LDT gene, (ii) deletions of the *erfK*, *ynhG*, *ycfS*, and *ybiS* LDT genes, (iii) a 13 bp deletion lowering the translation level of the gene encoding the isoleucyl-tRNA synthase (IleRS) thereby increasing synthesis of the (p) ppGpp alarmone, and (iv) an insertion of IS*1* in gene *sltY* encoding lytic transglycosylase Slt70 thereby promoting growth in the presence of ampicillin in liquid media (*Voedts et al., 2021*). Upon induction of *ycbB* by IPTG, *E. coli* M1.5 expresses high-level resistance to ampicillin and ceftriaxone following production of YcbB. Deletion of the *erfK*, *ycfS*, and *ybiS* genes abolishes the anchoring of the Braun lipoprotein and simplifies the *rp*HPLC muropeptide profile since stem peptides substituted by the Lys-Arg moiety of the lipoprotein are absent (*Magnet et al., 2007*).

The *ampG* permease gene (*Johnson et al., 2013*) of *E. coli* BW25113 M1.5 was deleted using the two-step procedure described by *Datsenko and Wanner, 2000*. The kanamycin resistance gene used for allelic exchange was amplified with oligonucleotides 5'-CCAAAAACTATCGTGCCAGC and 5'-A GAGTGACAACTGGGTGATG.

*E. coli Δ6ldt* is a derivative of strain BW25113 obtained by deletion of the six LDT genes present in *E. coli* (*erfK, ynhG, ycfS, ybiS, yafK,* and *ycbB*) (**Magnet et al., 2007**; **Magnet et al., 2008**; **Montón Silva et al., 2018**; **Morè et al., 2019**).

## Growth conditions and determination of the generation time

*E. coli* strains were grown in M9 minimal medium composed of 2 mM $MgSO_4$, 0.00025% $FeSO_4$, 12.8 g/L $Na_2HPO_4$, 3 g/L $KH_2PO_4$, 0.5 g/L NaCl, 1 g/L $NH_4Cl$, 1 µg/L thiamine, 1 g/L glucose (unlabeled M9 minimal medium). The labeled M9 minimal medium had the same composition except for the replacement of $NH_4Cl$ and glucose by [$^{15}$N]$NH_4Cl$ and uniformly labeled [$^{13}$C]glucose at the same concentrations (99% labeling for both compounds; Cambridge Isotope Laboratories). The growth media were sterilized by filtration (Millex-LG 0.2 µm, Merck Millipore).

Bacteria were grown on brain heart infusion agar plates (Difco) to initiate the experiment with an isolated colony that was inoculated in 25 mL of labeled minimal medium. The preculture was incubated for 18 hr at 37°C with aeration (180 rpm). The entire preculture (25 mL) was inoculated into 1 L of labeled M9 medium and the resulting culture was incubated until the optical density at 600 nm ($OD_{600}$) reached 0.4. Thus, both the preculture and the culture were performed in labeled liquid medium to avoid isotopic dilution. Bacteria were collected by centrifugation (8200× *g*, 7 min, 22°C), resuspended in 1 L of prewarmed (37°C) unlabeled M9 medium, and incubation was continued in the same conditions. Samples (200 mL) were withdrawn at various times and bacteria were collected by centrifugation (17,000× *g*, 5 min, 4°C) and stored at –55°C.

The generation time was deduced from the inverse of the slope of semi-logarithmic plots (log of $OD_{600}$ as a function of time).

For microscopy analyses, 1 mL of the culture used for peptidoglycan analysis was withdrawn and fixed by addition of paraformaldehyde (final concentration 4%; v/v). The sample was incubated at 4°C for 15 min, centrifuged, the pellet was washed twice with 1 mL of phosphate buffer saline (PBS), and resuspended in 1 mL of PBS. Bacteria were imaged using agarose-pad slices (1% agarose in PBS). Phase contrast images of bacteria were taken using a 100× magnification objective on a Nikon TI inverted microscope.

## Peptidoglycan preparation and purification of muropeptides

Peptidoglycan was extracted by the hot SDS procedure and treated with pronase and trypsin (**Arbeloa et al., 2004**). Muropeptides were solubilized by digestion with lysozyme and mutanolysin, reduced with $NaBH_4$, and separated by *rp*HPLC on a $C_{18}$ column (Hypersil GOLD aQ 250 × 4.6, 3 µm; ThermoFisher) at a flow rate of 0.7 mL/min. A linear gradient (0–100%) was applied between 11.1 and 105.2 min at room temperature (buffer A: 0.1% TFA; buffer B: 0.1% TFA, 20% acetonitrile; v/v). The absorbance was monitored at 205 nm and peaks corresponding to the major monomers and dimers were individually collected, lyophilized, solubilized in 30 µL of water, and stored at –20°C.

## Calculation of the molar ratios of muropeptides

*rp*HPLC analysis of sacculi digested by muramidases provided relative areas for peaks detected by the absorbance at 205 nm. In order to evaluate the relative molar abundance of muropeptides, we used the correction factors reported by Glauner that take into account the number of disaccharide units, amide bonds, 1,6-anhydro-ends, and Lys and Arg residues (**Glauner, 1988**). Since the relative amount of the muropeptides did not vary upon growth (i.e. between samples collected at various times following the medium switch), or between samples from independent experiments (i.e. between biological repeats obtained for the same strain in the same growth conditions), an average molar composition of the sacculi was deduced from the *rp*HPLC chromatograms and the correction factors reported by **Glauner, 1988**.

## Mass spectrometry

For routine identification of muropeptides, samples were diluted (10 µL plus 50 µL of water) and 2 µL were injected into the mass spectrometer (Maxis II ETD, Bruker, France) at a flow rate of 0.1 mL/min (50% acetonitrile, 50% water; v/v acidified with 0.1% formic acid; v/v). Mass spectra were acquired in the positive mode with a capillary voltage of 3500 V, a pre-storage pulse of 18 µs, ion funnel 1 RF 300 Vp-p. The *m/z* scan range was from 300 to 1850 at a speed of 2 Hz. Transfer time stepping was

enabled with the following parameters: RF values 400 and 1200 Vp-p, transfer times 30 and 90 µs, timing 50% and 50%. MS/MS spectra were obtained using a collision energy of 50 eV in the *m/z* range of 150–1000 for muropeptide monomers and 150–2000 for muropeptide dimers with isolation width of 1. The applied collision energy was varied between 77% and 100% of the 50 eV setting with timings of 33% and 67%, respectively.

The structural characterization of the labeled GlcNAc-MurNAc-tetrapeptide isotopologues reported in *Table 1* was performed on a ThermoFisher Orbitrap Fusion Lumos instrument operated in nanospray mode. The 50% acetonitrile in water analyte solution was acidified to 1% formic acid and introduced in the instrument using conventional nanospray glass emitter tips. The source voltage was set to 3000–3500 V and the sample was introduced without pneumatic assistance. The ions under each one of the peaks that made the isotopic cluster were fragmented individually by tune-mode quadrupole-selection of the corresponding ions in a 0.4 *m/z*-width selection window centered on the peak apex. That narrow selection window proved to be selective enough to only fragment the ions under each isotopic cluster peak of interest (from most intense to less intense: *m/z* 986.52, *m/z* 985.51, *m/z* 984.51). The fragmentation was obtained for a collision-induced dissociation energy of 10 eV. The MS/MS data were acquired in 200–1000 *m/z* range spectra (at least 50 MS/MS scans were acquired). Raw MS/MS data files were converted to the standard mzML file format using ProteoWizard's msconvert tool (*Adusumilli and Mallick, 2017*). The obtained data files were scrutinized using the mineXpert2 software program (*Langella and Rusconi, 2021*).

The software developments required to predict and analyze the labeled/unlabeled muropeptide ions isotopic clusters either in MS or MS/MS experiments are hosted at https://gitlab.com/kantund-peterpan/masseltof; *Atze, 2021* and published under a Free Software license.

## Acknowledgements

Funding: *Agence Nationale de la Recherche* (ANR-16-CE11-0030-12, TransPepNMR). *National Institute of Allergy and Infectious Diseases* (grant R56AI045626). Routine MS experiments were carried over at the *Plateau Technique de Spectrométrie de Masse Bio-organique* of the *Muséum national d'Histoire naturelle*.

---

## Additional information

### Funding

| Funder | Grant reference number | Author |
| --- | --- | --- |
| Agence Nationale de la Recherche | ANR-16-CE11-0030-12 | Heiner Atze Michel Arthur |
| National Institute of Allergy and Infectious Diseases | R56AI045626 | Yucheng Liang |

The funders had no role in study design, data collection and interpretation, or the decision to submit the work for publication.

### Author contributions

Heiner Atze, Data curation, Formal analysis, Investigation, Methodology, Software, Writing – original draft; Yucheng Liang, Conceptualization, Data curation, Formal analysis, Methodology, Software, Writing – review and editing; Jean-Emmanuel Hugonnet, Conceptualization, Formal analysis, Funding acquisition, Investigation, Methodology, Project administration, Supervision, Validation, Writing – original draft, Writing – review and editing; Arnaud Gutierrez, Data curation, Writing – review and editing; Filippo Rusconi, Conceptualization, Data curation, Formal analysis, Funding acquisition, Investigation, Methodology, Project administration, Software, Supervision, Validation, Writing – original draft, Writing – review and editing; Michel Arthur, Conceptualization, Data curation, Formal analysis, Funding acquisition, Investigation, Methodology, Project administration, Supervision, Validation, Writing – original draft, Writing – review and editing

### Author ORCIDs
Heiner Atze http://orcid.org/0000-0003-1497-6373
Jean-Emmanuel Hugonnet http://orcid.org/0000-0003-4150-0944
Filippo Rusconi http://orcid.org/0000-0003-1822-0397
Michel Arthur http://orcid.org/0000-0003-1007-636X

### Decision letter and Author response
Decision letter https://doi.org/10.7554/eLife.72863.sa1
Author response https://doi.org/10.7554/eLife.72863.sa2

---

## Additional files

### Supplementary files
• Supplementary file 1. Supplemental Tables. (a) Muropeptide composition of the peptidoglycan of strains BW25113Δ6*ldt*, M1.5, and BW25113 'wild-type'. [a]Data are the mean ± standard deviation from independent experiments (see note c). [b]β-Lactam added to the growth medium. AZT, aztreonam at 12 µg/mL; MEC, mecillinam at 2.5 µg/mL; AMP, ampicillin at 16 µg/mL. [c]The number of independent peptidoglycan analyses is indicated in parenthesis. Abbreviations: ND, not detected; Tri, tripeptide monomer; Tetra, tetrapeptide monomer; Penta, pentapeptide monomer; Tri→Tri, Tri→Tetra, and Tri→Penta, 3→3 cross-linked dimers; Tetra→Tri and Tetra→Tetra, 4→3 cross-linked dimers. The inter-peptide cross-link direction is indicated using the donor→acceptor conventional notation. (b) Generation time for growth of BW25113 derivatives in M1 minimal medium [a]β-lactam added to the growth medium. AZT, aztreonam at 12 µg/mL; MEC, mecillinam at 2.5 µg/mL; AMP, ampicillin at 16 µg/mL. [b]For growth in the presence of aztreonam and mecillinam, the values were deduced from the variation in $OD_{600}$ during 90 min. (c) Susceptibility of *Escherichia coli* strains to β-lactams. Data are the medians from six experiments.

• Supplementary file 2. Tandem mass spectrometry analysis of muropeptides.

• Supplementary file 3. Supplementary Material and Methods.

• Transparent reporting form

### Data availability
MS/MS spectra have been provided in Supplementary data file. The software developments required to predict and analyze the labeled/unlabeled muropeptide ions isotopic clusters either in MS or MS/MS experiments are hosted at https://gitlab.com/kantundpeterpan/masseltof, (copy archived at swh:1:rev:c45a7e66a8a89c436d41b8431938b35acfa4e53b) and published under a Free Software license.

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
