## [Editor Report]

The authors describe the innovative use of a heavy-isotope labeling strategy combined with mass spectrometry analysis to investigate the role of peptidoglycan biosynthesis by an L,D-transpeptidase and penicillin-binding proteins in *Escherichia coli*. They use isotopic labelling of peptidoglycan followed by a chase experiment with labels to study how new subunits are assembled into the pre-existing sacculus. The data suggests that new material is inserted one strand at a time on the lateral wall while it appears to be inserted as multiple strands at the division septum. The data are novel and provide important insights, together with notable methodological advances.

---

## [Decision Letter]

**Decision letter after peer review:**

Thank you for submitting your article "Heavy isotope labeling and mass spectrometry reveal unexpected remodeling of bacterial cell wall expansion in response to drugs" for consideration by *eLife*. Your article has been reviewed by 3 peer reviewers, and the evaluation has been overseen by Bavesh Kana as the Senior and Reviewing Editor. The following individual involved in review of your submission has agreed to reveal their identity: Tanneke den Blaauwen (Reviewer #3).

Essential revisions:

1. While the data on the one-by-one strand insertion in the lateral wall is supported by all the presented data, there seems to be a conflict in the data presented for the insertion of new peptidoglycan strands during cell division. The calculation of the ratio of new vs old peptidoglycan presented in Supplementary figure 12 for bacteria treated with mecillinam (that grow as cocci and perform only cell division) indeed suggests that multiple peptidoglycan strands are inserted from the beginning of the chase. However, this data is based on abundance of labeled, hybrid and unlabeled peptidoglycan fragments presumably presented in Figure 8. The data on panel D, left graph does not show any immediate presence in the peptidoglycan of unlabeled peptidoglycan fragments. In fact, these are as low in abundance as in none treated or aztreonam treated cells (that elongate but do not divide). These two figures are hard to reconcile.

2. As indicated in several figure legends, the data presented is from two independent experiments. Does this mean that the authors only performed twice the chase experiments? The experiment itself is rather simple to perform and it's the downstream analysis that is extremely time consuming. How confident are the authors of the reproducibility of the data?

3. The cells are grown in mecillinam and aztreonam, which inhibit length growth and division, respectively. The cells are expected to have a round and filamentous morphology, respectively, but this is not shown in the manuscript, where a very "non wild-type" strain is used for the experiments. The concentration of the antibiotics is perhaps based on the literature, but maybe this strain has difference susceptibility to the antibiotics? Using the wild type strain in these experiments would be important to demonstrate relevance of the findings.

4. In the absence of LD-TPases the insertion of new PG is predominantly new-old for the first 20 minutes, whereas in the M1.5 strain, this is 50% new-old and 50% new-new. Their mass doubling times are 67 and 90 min, respectively. Both strains should contain dividing cells directly after the medium change. If septum formation is multistrand insertion, one would expect new-new in both strains. Then aztreonam is added to the δ 6 LD-TPases strain and the mode of insertion remains the same (new-old), suggesting that length growth is single strand insertion. However, this cannot be concluded because with division it was also single stranded. Then in the presence of mecillinam the authors claim that they see new-old and new-new insertion and therefore, claim that division requires multiple strand insertion. However, in the graph of the mecillinam cells, (1) it looks as if the PG synthesis is considerably slowed down in comparison to the other two situations. This is unexpected given the that the mass doubling time did not change according to Table S2. (2) If one extrapolates the lines in the graph, they are not that different from the strains without antibiotics or with aztreonam. (3) The new-old dominates, which one does not expect from cells that enlarge predominantly through division. The explanation that the futile cycle of PG TGase activity may result in insertion of old-new strand by PBP1b and 1a is plausible, but it does not prove that septation is multistrand. Would it be possible, to calculate the amount of surface generated through septation and elongation per min and predict the expected increase in old-new and new-new in the M1.5 strain to see if the observed data would fit the proposed model?

5. In general, it is a pity that no wild-type strain is used for the new method. The conclusion of the paper is very important, i. e. new PG is inserted strand by strand likely by the combined action of an TPase and an endopeptidase. The absence of data on a WT strain diminishes the soundness of the conclusion. Since it can be expected that this paper will be cited for a long time by many people, it is important that it is complete. Experiments with the wild type should be reported.

6. The discrimination of labelled versus unlabelled assumes that if one isotope is present, the molecule is considered to be labelled and if no isotopes are present the molecule is unlabelled. A situation must exist (as is outlined in Figure S14 and how the mass spectra data are poled in S15) in which for instance glucosamine is made from a mixture of labelled and unlabelled 13C and existing labelled DAP (which was also reported as having a considerable cellular pool) is coupled to an newly synthesized (unlabeled) D-Ala-D-Ala. This would result in a high number of different MS peaks which all present the same structure. Assuming that the relative abundance of these variable labelled molecules will be the same for all muropeptides, but H1 and H2, is 5 min enough to cause every newly synthesized muropeptide to be fully unlabelled during the chase or is there a mixture for some time and how longs do these mixtures last? Maybe this gives some information on the presence of the pools of the various parts of the muropeptides?

7. The GlcNAc-anhydro-MurNAc- peptides must have a discrete mass from the standard muropeptides, but are not identified as such. Why not? It gives information in the length of the glycan strands and as such is interesting.

8. Page 20: "Thus, the absence of old-new 3-3 cross-linked Tri-Tetra dimer indicates that YcbB discriminates neo-synthesized tetrapeptide stems (used as donors) from existing tetrapeptide stems (used as acceptors)" Could it be that these crosslinks were below the detection limit? It is important because If YcbB is indeed only able to make 3-3 crosslinks from newly synthesized PG, this implies that repair is only possible if a TGase polymerizes PG. It has been shown by Morè et al., (Morè, N., Martorana, A.M., Biboy, J., Otten, C., Winkle, M., Serrano, C.K.G., et al. (2019) Peptidoglycan Remodeling Enables *Escherichia coli* To Survive Severe Outer Membrane Assembly Defect. mBio 10.) that YcbB works together with PBP1B and PBP6a and the absence of old-new may explain and support this further.

Other corrections:

1. In the introduction, the authors mention "lytic glycosyltransferases". The term "lytic transglycosylase" seems more appropriate.

2. Ln 62: a comma is missing after "in *E. coli*".

3. Ln 104: Change "NMR" to "Solution-state NMR" because solid-state NMR, although limited in its applications, has been successful in characterizing the tertiary structure of peptidoglycan in bacteria. For example, "*Staphylococcus aureus* peptidoglycan tertiary structure from carbon-13 spin diffusion" (JACS, 2009).

4. The Figure S2. ADRR calculation assumes the PG model which consists primarily of PG dimers with 50% crosslinking efficiency. The literature is in good agreement with an average PG crosslinking efficiency in *E. coli* observed at 50% (ranging from 30~60% depending on the strains),1 but in addition to a significant number of monomers (30 to 55%) and dimers (41 to 51%), there are trimers (4 to 10%) and tetramers (less than 1%) that are found in *E. coli*.2 These oligomeric muropeptide species are low in their abundance, however, a statement on how these oligomers are consistent with the insertion models would be important.

5. Ln 211: insert ", Peak A" after "The second peak of the isotopic cluster".

6. Ln 224: Change "cluster D" to "Peak D".

7. Ln 227: Change "clusters" to "Peaks".

8. Ln 231: "(99% for each isotope)" mentioned in the text is for the isotopic enrichments for the heavy-labeled glucose and ammonium chloride. However, the isotope incorporation efficiency in *E. coli* seems to be less than 99% because of the observed spectrum of Figure 2D which shows the larger peak intensities for Peaks A, B, and C compared to the corresponding ratio shown in the simulated spectrum (Figure 2E). Since isotopic distribution is highly sensitive to dilution, perhaps 25 ml of preculture used to inoculate 1L of labeled media was grown in unlabeled M9. If so, then the dilution is approximately 25ml/(100ml + 25ml) = 2.4% and thus the expected isotopic incorporation efficiency is 96.5% (97.5% * 99%) which may account for the differences in the observed and calculated Peaks for A, B, and C? This is a minor point since it does not change the overall analysis.

9. Ln 251, Ln 264: "Table 2" is missing from the manuscript.

10. Ln 308: insert "heavy-isotope labeled" before "minimal medium".

11. Ln 467 "Tri-Tri and Tetra-Tri dimers formed by PBPs and YcbB, respectively" should be changed to "Tri-Tri and Tetra-Tri dimers formed by YcbB and PBPs, respectively".

12. Ln 468: "were almost exclusively of the new-old type (Figure 7)" But Figure 7 A (Tri-Tri) and B (Tetra-Tri) show an even mixture of both new-old and new-new. Hence, the statement "exclusively of new-old" is not entirely correct.

13. Ln 468-469: "In contrast, Tri-Tetra and Tetra-Tetra dimers formed by these enzymes were a mixture of new-new and new-old" but the figure shows only new-new.

14. Ln 475: The statement "Thus, the mode of insertion of PG subunits depended upon the structure of the acceptor stem … rather than upon the type of transpeptidase" is based on the observation from Figure 7 which shows that panels A and B (Tri as acceptor), and panels C and D (Tetra as acceptor) showed similar kinetic behaviors. The addition of a few sentences for an explanation would add clarity to the statement.

15. Figure 7 provides insight into the flux of how much new (unlabeled) PG is incorporated. For example, in Figure 7 C (Tri-Tetra 3-3 CL) at 85 mins the observed N-N is at ~80%, O-O at ~10%, and N-O at ~10%, then the relative total amount of Old PG subunit (2*10% + 10% = 30%) to the New PG subunit (2*80% + 10% = 170%) for Tri-Tetra of 3-3 CL is 3:17. The approximate ratio of old to new for Tetra-Tetra of 4-3 is approximately 1:19. This indicates that the newly formed PG subunits (nascent PG) are incorporated into both 4-3 and 3-3 CL with a tetrapeptide stem, although more actively to 4-3 CL. In contrast, the approximate ratio of old to new for Tri-Tri (3-3 CL) and Tetra-Tri (4-3) is 9:17 and 1:2, respectively. This indicates that the newly formed PG subunit in the form of a tripeptide acceptor is not readily incorporated into the cell wall. Hence, the insertion into PG shows dependence on the PG stem structure of the acceptor. Alternatively, the concentrations of Tri and Tetra are not independent but coupled, most likely high concentration of Tetrapeptide PG stems (both monomers and dimers) as shown Figure 6 (b-f) is likely to be the determinant for the observed kinetics.

16. Ln 519: Please explain the amounts of antibiotics that were added, Mec (2.5 ug/ml) and AZT (12 ug/ml), to *E. coli* before the medium switch.

17. Ln 557 Figure 9A: "… grown in the presence of ampicillin", at what time point was the sample collected?

18. Ln 546-549: "New-new rather than new-old isotopologues were predominantly detected among dimers……. In contrast, the Tri-Tetra dimers in the PG of M1.5 grown in the absence of ampicillin mainly contained the new-old isotopologues (Figure 7)". Figures below show kinetics for Tri-Tetra dimers from Figure 9 (left) and Figure 7C (right). Contrary to the statement, PG of M1.5 grown in the absence of ampicillin (Figure 7C, right) contained a significant amount of new-new isotopologues.

19. For Figures 5, 7, 8, and 9, please provide which program was used to fit the data. Also, what was the kinetic model used for the fit? Does the model assume sequential event of A to B to C (conversion from old-old to new-old and then to new-new)? An explanation for a buildup of new-old in Tri-Tri but not in Tri-Tetra in *E. coli* M1.5 grown in the presence of ampicillin (as shown in Figure 9C below) would be helpful.

20. Ln 577-579: The rationale for switching the medium from labeled to unlabeled was that it "facilitates the detection of neo-synthesized isotopologues present at a low abundance at early times after the medium switch." It was not clear how this facilitates the detection of neo-synthesized isotopologues compared to switching from unlabeled to label medium. Additional sentences explaining this would be of interest to the readers.

21. Ln 610: "old-new" should be changed to "old-old".

22. Ln 671-673: Confusing statement. Ln 672: "new-old" should be "new-new"?

23. Figure S12 is of great significance and thus recommend moving this as one of the main figures in the text.

24. Page 20 "Kinetic analyses revealed that the Tri-Tri and Tetra-Tri dimers formed by PBPs and YcbB, respectively, were almost exclusively of the new-old type (Figure 7). Should these be YcbB and PBPs, respectively.

25. In Figure 6, S7, S8 and S9 strain M1.4 is used, but this strain is not described in the method section.

26. The *E. coli* BW25113 rph mutation is a frame shift that leads to pyrimidine starvation. To have an optimal DNA replication cycle in M9 medium it needs some supplements, which were not added according to the Materials and methods. This may not affect the PG mode of insertion but perhaps the authors can keep this in mind for future experiments? All strains used in this study are derivatives of this strain. *E. coli* BW25113 [Δ(araD-araB)567 Δ(rhaD-rhaB)568 ΔlacZ4787 (::rrnB-3) hsdR514 rph-1, lacI+] BW25113 Needs uracil 20 ug/ml, when grown in minimal medium. KF Jensen J Bacteriol. 1993 Jun;175(11):3401-7. Grenier, F., Matteau, D., Baby, V., & Rodrigue, S. (2014). Complete Genome Sequence of *Escherichia coli* BW25113. Genome Announcements, 2(5). doi:10.1128/genomeA.01038-14. Needs also thymidine, 2 ug/ml according to this paper. J Bacteriol. 1998 Jun;180(11):2992-4. Determining the optimal thymidine concentration for growing Thy- *Escherichia coli* strains. Molina F1, Jiménez-Sánchez A, Guzmán EC.

---

## [Author Response]

Essential revisions:1. While the data on the one-by-one strand insertion in the lateral wall is supported by all the presented data, there seems to be a conflict in the data presented for the insertion of new peptidoglycan strands during cell division. The calculation of the ratio of new vs old peptidoglycan presented in Supplementary figure 12 for bacteria treated with mecillinam (that grow as cocci and perform only cell division) indeed suggests that multiple peptidoglycan strands are inserted from the beginning of the chase. However, this data is based on abundance of labeled, hybrid and unlabeled peptidoglycan fragments presumably presented in Figure 8. The data on panel D, left graph does not show any immediate presence in the peptidoglycan of unlabeled peptidoglycan fragments. In fact, these are as low in abundance as in none treated or aztreonam treated cells (that elongate but do not divide). These two figures are hard to reconcile.

The data presented in Figure 8 and Supplementary Figure S12 (Figure 11 in the revised manuscript) are extracted from the same experiments but cannot be compared as suggested by the reviewer since they concern the kinetics of variations in the percentage of the various isotopologues (new→new, new→old, and old→old) *versus* the ADRNS, respectively. The latter, ADRNS, provides an estimate of the acceptor-to-donor ratio in neo-synthesized stems in dimers. It therefore does not depend upon the absolute rate of replacement of old (heavy) by new (light) stems. Instead, it provides an estimate of the proportion of new and old stems in the acceptor and donor position of dimers. The data appearing in the two figures are correct: Even though the absolute rate of replacement of old stems by new ones is low in the presence of mecillinam (Figure 8D; right panel), it is clear that the old→new and new→new dimers both increase at the beginning of the kinetics resulting in an ADRNS of 0.31 at 4 min (Figure 11). In contrast, the appearance of new→new dimers is delayed in the presence of aztreonam resulting in a lower ADRNS (0.14 at 5 min; Figure 11; extrapolation of 0.10 at t = 0). In the presence of aztreonam, the ADRNS values increase with time as expected from multiple and independent rounds of insertion of new material at the same location. We added three sentences in the legend to Figure 11 to indicate that data presented in this figure originate from the treatment of data appearing in Figure 8. These sentences also explain the origin of the apparent discrepancy underscored by the reviewer.

Additional sentences appearing in the legend to Figure 11 are as follows:

“Curves are obtained by linear regression. ADRNS were deduced from the relative abundance of new→old and new→new dimers appearing in Figure 8. Please note that the ADRNS was the highest for growth in the presence of mecillinam (this figure) in spite of the fact that the overall rate of replacement of old stems by new stems was the lowest for this growth condition, as evidenced by the slow rise of both the new→old and new→new dimers (Figure 8). This apparent discrepancy is accounted for by the fact that the ADRNS is a ratio solely based on neo-synthesized dimers.”

2. As indicated in several figure legends, the data presented is from two independent experiments. Does this mean that the authors only performed twice the chase experiments? The experiment itself is rather simple to perform and it's the downstream analysis that is extremely time consuming. How confident are the authors of the reproducibility of the data?

The reviewer is correct in stating that the experiments were performed twice. However, each one of the two experiments includes an independent analysis of at least five biological samples recovered at various time points (peptidoglycan extractions, digestion with muramidases, separation of the muropeptides by *rp*HPLC, and isotopologue identification by mass spectrometry). This approach provides evidence that the determination of muropeptide isotopologues between data points from the same experiments and between two fully independent experiments are consistent and reproducible. Our strategy was to focus on extensive characterization of a limited number of samples rather than generating a larger set of less extensively characterized data. Nonetheless, in the additional experiment addressing the next comment (comment 3), we generated a set of four biological repeats and the reviewers will find an analysis of the isotopologue measurement variability added to our answers as a spreadsheet file entitled “Variability analysis, answer to comment 2”.

3. The cells are grown in mecillinam and aztreonam, which inhibit length growth and division, respectively. The cells are expected to have a round and filamentous morphology, respectively, but this is not shown in the manuscript, where a very "non wild-type" strain is used for the experiments. The concentration of the antibiotics is perhaps based on the literature, but maybe this strain has difference susceptibility to the antibiotics? Using the wild type strain in these experiments would be important to demonstrate relevance of the findings.

We determined the minimal inhibitory concentrations (MICs) of aztreonam and mecillinam against strains BW25113 and BW25113Δ6*ldt* (additional Supplementary Table S3). Aztreonam and mecillinam were used at concentrations (2.5 µg/mL and 12 µg/mL, respectively) that were thus inhibitory for both strains (as we initially expected).

Strain BW25113Δ6*ldt* does not produce any L,D-transpeptidase. In strain BW25113, these enzymes marginally contribute to peptidoglycan cross-linking since YcbB and (p)ppGpp are produced at “wild-type” levels (Hugonnet *et al.* 2016). Thus, the MICs were expected to be similar for both strains and were experimentally shown to be so in the additional analysis presented in Table S3 (introduced Lines 481 to 483 of the revised manuscript):

“Aztreonam and mecillinam were used at inhibitory concentrations corresponding to 200- and 20-fold the MIC of the drugs, respectively (Supplementary Table S3).”

As requested by the reviewers, we performed a morphology analysis (additional Figure 8 – figure supplement 1). Mecillinam at 2.5 µg/mL and aztreonam at 12 µg/mL resulted in the expected changes in bacterial cell morphology, the formation of filaments or of spheres, respectively. The figure was introduced Lines 483-486 of the revised manuscript:

“Exposure to aztreonam or mecillinam resulted in the morphological changes expected for the inhibition of septum formation (filamentation) or of side wall synthesis (rounding), respectively (Figure 8 – figure supplement 1).”

The procedure to prepare the sample is described Lines 734 to 739 of the revised manuscript:

“For microscopy analyses, 1 mL of the culture used for peptidoglycan analysis was withdrawn and fixed by addition of paraformaldehyde (final concentration 4%; v/v). The sample was incubated at 4 °C for 15 min, centrifuged, the pellet was washed twice with 1 mL of phosphate buffer saline (PBS), and resuspended in 1 mL of PBS. Bacteria were imaged using agarose-pad slices (1% agarose in PBS). Phase contrast images of bacteria were taken using a 100x magnification objective on a Nikon TI inverted microscope.”

As recommended by the reviewers, we have repeated the full experiment with the parental “wild-type” strain BW25113 in the absence of β-lactam and in the presence of aztreonam or mecillinam. The results appear in Figure 8 – figure supplement 2. The initial choice of the BW25113Δ6*ldt* strain was motivated by the highly simplified muropeptide profile of this strain (absence of tripeptide and of 3→3 cross-linked dimers) that was essential to extensively characterize all the cross-linked muropeptides. We maintain that this choice was justified and additionally provides the requested analysis of a “wild-type” strain, *i*.*e*. the parental strain used for construction of BW25113Δ6*ldt*, which does not harbor any known mutation in genes involved in peptidoglycan metabolism. As expected from the similarity between the muropeptide compositions of strains BW25113 and BW25113Δ6*ldt*, the ADRNS values were similar for BW25113Δ6*ldt* and parental “wild-type” strain BW25113 (Figure 8 – figure supplement 3). This additional experiment establishes that the results obtained with BW25113Δ6*ldt* (in the first version of the manuscript) were not biased by the absence of the six *ldt* genes.

The analyses performed with the parent “wild-type” strain grown in the absence of β-lactams or in the presence of aztreonam or mecillinam appear in an additional section at the end of the results (Lines 506 to 522):

**“**Mode of insertion of peptidoglycan subunits in a bacterial strain harboring the wild-type complement of PBP and L,D-transpeptidase genes. Our last objective was to characterize the mode of peptidoglycan synthesis in a wild-type background with respect to genes involved in peptidoglycan synthesis. Toward this aim, we analyzed the peptidoglycan from the parental “wild-type” strain BW25113. Figure 8- figure supplement 2 provides a comparison of the kinetics obtained for both the BW25113∆6*ldt* and parental BW25113 strains grown in the absence of β-lactams and in the presence of aztreonam or mecillinam. This analysis showed that the conclusions drawn from the analyses performed with strain BW25113Δ6*ldt* (Figure 8) are not biased by the deletion of the six *ldt* genes. The ADRNS values were similar for both strains (Figure 8- figure supplement 3) indicating that the mode of insertion of new stems in the Tetra→Tetra dimers is not affected by the deletion of the six *ldt* genes. For the sake of completeness, we also compared the kinetics of appearance of muropeptides resulting from the activity of L,D-transpeptidases that are present in low amount in the parental strain BW25113 and are more abundant in mutant M1.5 due to overproduction of YcbB (Supplementary Table S1). Qualitatively, the kinetics were similar for muropeptides extracted from these two strains (Figure 8- figure supplement 4). Overall, the replacement of old by new muropeptides was more rapid for the “wild type” strain than for mutant M1.5.”

4. In the absence of LD-TPases the insertion of new PG is predominantly new-old for the first 20 minutes, whereas in the M1.5 strain, this is 50% new-old and 50% new-new. Their mass doubling times are 67 and 90 min, respectively. Both strains should contain dividing cells directly after the medium change. If septum formation is multistrand insertion, one would expect new-new in both strains. Then aztreonam is added to the δ 6 LD-TPases strain and the mode of insertion remains the same (new-old), suggesting that length growth is single strand insertion. However, this cannot be concluded because with division it was also single stranded. Then in the presence of mecillinam the authors claim that they see new-old and new-new insertion and therefore, claim that division requires multiple strand insertion. However, in the graph of the mecillinam cells, (1) it looks as if the PG synthesis is considerably slowed down in comparison to the other two situations. This is unexpected given the that the mass doubling time did not change according to Table S2. (2) If one extrapolates the lines in the graph, they are not that different from the strains without antibiotics or with aztreonam. (3) The new-old dominates, which one does not expect from cells that enlarge predominantly through division. The explanation that the futile cycle of PG TGase activity may result in insertion of old-new strand by PBP1b and 1a is plausible, but it does not prove that septation is multistrand. Would it be possible, to calculate the amount of surface generated through septation and elongation per min and predict the expected increase in old-new and new-new in the M1.5 strain to see if the observed data would fit the proposed model?

As indicated in our answers to comment #2, our data actually do show, based on ADRNS determination, that the mode of insertion is different in the presence of aztreonam (mostly single strand) and in the presence of mecillinam (a combination of single and multi-strand insertion). Our answers to the additional points raised in comment #4 are as follows:

(i) The author states that “it looks as if the PG synthesis is considerably slowed down [in the presence of mecillinam] in comparison to the other two situations. This is unexpected given the that the mass doubling time did not change according to Table S2.”

The analysis proposed by the reviewer should take into account the relative contributions of cell poles and side walls to the cell surface in addition to overall growth. If the length of the side wall in the long axis of bacterial cells is twice the cell diameter, then the contribution of the poles and side walls to the cell surface is 33% and 67%, respectively. It is thus expected that the overall synthesis of new material is slowed down to a greater extent by blocking elongation of the side wall than by blocking formation of the poles. The kinetics observed in the absence of the drugs can be accounted for by an additive combination of kinetics obtained in the presence of mecillinam and aztreonam with a greater contribution of the latter condition (extension of side walls). Regarding the variation in OD, it is not obvious that the increase in the cell surface is directly proportional to the optical density. In the legend to Table S2 (first version of the manuscript, footnote b), we have clearly indicated that generation times for the aztreonam and mecillinam conditions were deduced from the variation in the OD_600_ during 90 min. We think that these data should appear in the manuscript as an estimate of potential increases in the biomass. Obviously, these estimates do not reflect steady state growth. Thus, they should not be over-interpreted to relate changes in peptidoglycan composition to putative growth rates that cannot be precisely defined for all growth conditions used in this study.

2) “If one extrapolates the lines in the graph, they are not that different from the strains without antibiotics or with aztreonam.”

As indicated in the first version of the manuscript, extrapolation should be made for t=0 min (Figure 11; revised version, formerly Figure S12). When the proportion of new material increases in the wall, the probability that new stems are cross-linked to unlabeled material inserted after the medium switch increases. It is therefore expected that extrapolations for long incubation times can be convergent for different modes of insertion (single *versus* multiple strands) as observed at 40 min in Figure 11.

3. “The new-old dominates, which one does not expect from cells that enlarge predominantly through division. The explanation that the futile cycle of PG TGase activity may result in insertion of old-new strand by PBP1b and 1a is plausible, but it does not prove that septation is multistrand. Would it be possible, to calculate the amount of surface generated through septation and elongation per min and predict the expected increase in old-new and new-new in the M1.5 strain to see if the observed data would fit the proposed model?”

The proposed evaluation of the amount of surface generated through septation and elongation per min is feasible. However, we think that this analysis would be extremely speculative due to numerous confounding factors. (i) The rate of expansion of the cell surface may vary during the entire experiment and between individual bacterial cells. (ii) Steady state growth is unlikely to occur in the presence of β-lactam. (iii) Peptidoglycan recycling may be affected by exposure to drugs and is likely to differently affect the rate of synthesis of the septum and of side walls. For these reasons, we would like to stay with the initial conclusions that side wall synthesis mostly involves single strand insertion of new strands, whereas synthesis of the septum involves insertion of multiple strands. We think that we cannot provide a more refined estimate of the proportion of new material inserted as single *versus* multiple strand in bacteria treated with β-lactams.

5. In general, it is a pity that no wild-type strain is used for the new method. The conclusion of the paper is very important, i. e. new PG is inserted strand by strand likely by the combined action of an TPase and an endopeptidase. The absence of data on a WT strain diminishes the soundness of the conclusion. Since it can be expected that this paper will be cited for a long time by many people, it is important that it is complete. Experiments with the wild type should be reported.

As stated in our answer to comment #2, we have added to our study an extensive analysis of a “wild-type” strain, *i*.*e*. the parental strain BW25113, that does no harbor any known mutation directly affecting peptidoglycan synthesis. The results appear at the end of the Results section under the subheading “Mode of insertion of peptidoglycan subunits in a bacterial strain harboring the wild-type complement of PBP and L,D-transpeptidase genes” (Lines 506 to 522) and in three additional figures (Figure 8 – figure supplement 2, 3 and 4).

6. The discrimination of labelled versus unlabelled assumes that if one isotope is present, the molecule is considered to be labelled and if no isotopes are present the molecule is unlabelled. A situation must exist (as is outlined in Figure S14 and how the mass spectra data are poled in S15) in which for instance glucosamine is made from a mixture of labelled and unlabelled 13C and existing labelled DAP (which was also reported as having a considerable cellular pool) is coupled to an newly synthesized (unlabeled) D-Ala-D-Ala. This would result in a high number of different MS peaks which all present the same structure. Assuming that the relative abundance of these variable labelled molecules will be the same for all muropeptides, but H1 and H2, is 5 min enough to cause every newly synthesized muropeptide to be fully unlabelled during the chase or is there a mixture for some time and how longs do these mixtures last? Maybe this gives some information on the presence of the pools of the various parts of the muropeptides?

Our analyses are based on the characterization of hybrid muropeptides by tandem mass spectrometry. This analysis showed that the major peaks can be assigned to mosaic structures with a defined isotopic composition (*e*.*g*. Figure 4). A first conclusion is that the main isotopologues are made of combinations of labeled and unlabeled moieties (amino acids, glucosamine, acetate, and lactate) as opposed to a hybrid composition of labeled and unlabeled nuclei within these building blocks. Specifically, the fact that we only detected muropeptides with fully labeled or fully unlabeled glucosamine moieties indicates that glucosamine is not merely made of a mixture of heavy and light isotopes in agreement with the pathways for de novo synthesis of glucosamine and for the recycling of GlcNAc and MurNAc residues (Figure 1 – figure supplement 1). Hybrid h3 clearly originated from a pool of UDP-MurNAc-pentapeptide as expected from the abundance of this precursor in cytoplasm (Mengin-Lecreulx *et al.* 1982, cited in the first version of the manuscript). The h3 pool levels off over time, in contrast to h1 and h2, as expected from the exhaustion of a preexisting pool. The latter hybrids, h1 and h2, originate from recycling of existing (labeled) peptidoglycan. It is therefore expected that their accumulation will only level off when the proportion of old (labeled) peptidoglycan decreases due to insertion of new material. Regarding DAP, it is true that this precursor is also abundant. However, DAP is a precursor for L-lysine synthesis. Thus, the bulk of the DAP pool is used for protein synthesis. For D-Ala-D-Ala, we indeed detected hybrid tetrapeptides that were uniformly unlabeled except for the terminal D-Ala. We did not perform any pulse-chase experiment and could therefore not provide any insight into the size of the precursor pools and the fluxes in metabolic pathways.

7. The GlcNAc-anhydro-MurNAc- peptides must have a discrete mass from the standard muropeptides, but are not identified as such. Why not? It gives information in the length of the glycan strands and as such is interesting.

We agree that the determination of content of anhydro-MurNAc *versus* MurNAc-peptide could provide an indication of the length of glycan chains. However, in the context of our analysis, they are two caveats. First, the current determination of the isotopic composition of muropeptides relies on the detection of isotopologues with the same chemical composition as purified in a single peak of *rp*HPLC chromatograms. The determination of the proportion of the isotopologues is therefore extremely precise since it is not affected by the purification yields and the efficacy of ionization in the electrospray ion source. In contrast, estimates of the length of glycan chain relies on the determination of material present in different *rp*HPLC fractions. Our chromatographic gradient is optimized for MS analysis since is enables recovering muropeptides free from salt following lyophilization. However, the resolution of peaks is not sufficient for reliable determination of minor muropeptide species. Second, such analyses would only provide estimates of the average length of the glycan chains in a potentially heterogeneous material depending on the extents of peptidoglycan polymerization and recycling. Unfortunately, our study design does not allow us to adequately answer to the reviewer’s request. In addition, we think that discussing the length of glycan chains would introduce an entire new set of concepts in our study, which is already, we believe, highly complex for the readers. As shown in the figure below, the kinetics for muropeptides and anhydro-muropeptide were similar indicating that the behavior of the material at the end of the glycan chains does not substantially differ from that of the inner material. We would therefore like not to discuss anhydro-muropeptides in this study.

8. Page 20: "Thus, the absence of old-new 3-3 cross-linked Tri-Tetra dimer indicates that YcbB discriminates neo-synthesized tetrapeptide stems (used as donors) from existing tetrapeptide stems (used as acceptors)" Could it be that these crosslinks were below the detection limit? It is important because If YcbB is indeed only able to make 3-3 crosslinks from newly synthesized PG, this implies that repair is only possible if a TGase polymerizes PG. It has been shown by Morè et al., (Morè, N., Martorana, A.M., Biboy, J., Otten, C., Winkle, M., Serrano, C.K.G., et al. (2019) Peptidoglycan Remodeling Enables *Escherichia coli* To Survive Severe Outer Membrane Assembly Defect. mBio 10.) that YcbB works together with PBP1B and PBP6a and the absence of old-new may explain and support this further.

Thank you for drawing our attention to this interesting implication of our results. As proposed by the reviewer, we included a short paragraph on the required participation of the glycosyltransferase activity of PBP1b to peptidoglycan remodeling mediated by YcbB (Lines 452 to 461 of the revised manuscript):

“In this study, we analyzed the participation of YcbB to global peptidoglycan synthesis following overproduction of this L,D-transpeptidase and of the alarmone (p)ppGpp in combination with inhibition of PBPs in certain experiments. Previous work concluded that YcbB also participates in the repair of peptidoglycan in a wild-type background (Morè *et al.* 2019). In addition to YcbB, this process required glycosyltransferase and D,D-carboxypeptidase activities provided by PBP1b and PBP6a, respectively. This implies that peptidoglycan repair proceeds by polymerization of new glycan chains that are cross-linked by PBPs. It is worth noting that the mode of insertion of new peptidoglycan subunits delineated in the current study is fully compatible with this model since we detected only one type of 3→3 cross-linked dimers (new→old and not old→new).”

Other corrections:1. In the introduction, the authors mention "lytic glycosyltransferases". The term "lytic transglycosylase" seems more appropriate.

The text has been corrected as required (Line 124 of the revised manuscript).

2. Ln 62: a comma is missing after "in *E. coli*".

A comma has been added as required (Line 67 of the revised manuscript).

3. Ln 104: Change "NMR" to "Solution-state NMR" because solid-state NMR, although limited in its applications, has been successful in characterizing the tertiary structure of peptidoglycan in bacteria. For example, "*Staphylococcus aureus* peptidoglycan tertiary structure from carbon-13 spin diffusion" (JACS, 2009).

The reference cited in the first version of the manuscript (Kern et al., 2008) was also based on solid state NMR. We agree that the sentence in the first version of the manuscript was not sufficiently qualified. We propose to replace the sentence:

“NMR also failed to determine the structure of the polymer due to its heterogeneity and its important flexibility as measured by NMR relaxation (Kern et al., 2008).”

By the following sentence that incorporates the suggested citation:

“NMR spectroscopy also failed to determine the precise and complete structure of the polymer due to its heterogeneity and its important flexibility, as measured by solid state NMR relaxation (Kern et al., 2008, Sharif et al., 2009).”

Changes appear Lines 109 to 111 of the revised manuscript.

4. The Figure S2. ADRR calculation assumes the PG model which consists primarily of PG dimers with 50% crosslinking efficiency. The literature is in good agreement with an average PG crosslinking efficiency in *E. coli* observed at 50% (ranging from 30~60% depending on the strains),1 but in addition to a significant number of monomers (30 to 55%) and dimers (41 to 51%), there are trimers (4 to 10%) and tetramers (less than 1%) that are found in *E. coli*. 2 These oligomeric muropeptide species are low in their abundance, however, a statement on how these oligomers are consistent with the insertion models would be important.

We agree that trimers and tetramers contribute to the ADRR as calculated from data based on the previous radioactive labeling techniques. The situation is different for our study since the analysis of dimers is fully independent from that of trimers and tetramers. We advocate that including the analysis of trimers and tetramers is beyond the scope of the current study. In addition, structure assignment is hampered by the following issues: (i) The abundance of muropeptide trimers (7%) and tetramers (<1%) is too low to permit the required MS/MS analyses. (ii) For trimers and tetramers, all isomers are not resolved in individual *rp*HPLC peaks. This introduces ambiguity in the MS/MS analyses. (iii) The number of isotopologues for each isomer is very high, resulting in overlapping isotopic clusters. Thus, unambiguously assigning the structure of isotopologues is out of reach for molecules as complex as trimers and tetramers. We added a sentence Lines 285 to 288 of the revised manuscript to indicate that small amounts of trimers and tetramers were present in our preparations but that we were unable to characterize them: “The peptidoglycan preparations also contained small amounts of trimers and tetramers that were not analyzed in the current study because their low abundance, poor resolution by *rp*HPLC, and complex isotopologue composition precluded their full characterization by MS and MS/MS.”

5. Ln 211: insert ", Peak A" after "The second peak of the isotopic cluster".

The text has been modified as requested (Line 202 of the revised manuscript).

6. Ln 224: Change "cluster D" to "Peak D".

The text has been modified as requested (Line 214 of the revised manuscript).

7. Ln 227: Change "clusters" to "Peaks".

The text has been modified as requested (Line 217 of the revised manuscript).

8. Ln 231: "(99% for each isotope)" mentioned in the text is for the isotopic enrichments for the heavy-labeled glucose and ammonium chloride. However, the isotope incorporation efficiency in *E. coli* seems to be less than 99% because of the observed spectrum of Figure 2D which shows the larger peak intensities for Peaks A, B, and C compared to the corresponding ratio shown in the simulated spectrum (Figure 2E). Since isotopic distribution is highly sensitive to dilution, perhaps 25 ml of preculture used to inoculate 1L of labeled media was grown in unlabeled M9. If so, then the dilution is approximately 25ml/(100ml + 25ml) = 2.4% and thus the expected isotopic incorporation efficiency is 96.5% (97.5% * 99%) which may account for the differences in the observed and calculated Peaks for A, B, and C? This is a minor point since it does not change the overall analysis.

We were aware of the impact of the inoculum, if performed in the unlabeled medium, on the labeling efficacy. As described in the first version of the manuscript, both the preculture and culture were performed in the labeled medium to avoid any isotopic dilution. We added a short sentence in parenthesis to emphasis this fact in the “Results” section (Lines 222 and 223 of the revised manuscript): “please, note that both the preculture and the culture were performed in labeled medium to avoid isotopic dilution.” We also added a sentence in the “Materials and methods” section (Lines 726 to 727 of the revised manuscript): “Thus, both the preculture and the culture were performed in labeled liquid medium to avoid isotopic dilution.”

We agree that the chromatogram displayed in Figure 2E shows a slight enrichment in isotopologues containing one or two light isotopes in comparison to the simulated spectrum. Correcting the extent (99%) of labeling reported for ^13^C- and ^15^N-labeling of glucose and ammonium chloride by the manufacturer of these compounds could be used to improve the fit between simulated and observed spectra. However, introducing this modification, which is in the order of 0.2% (98,8% instead of 99%), will require complex description and justification, with minimal improvement the manuscript.

9. Ln 251, Ln 264: "Table 2" is missing from the manuscript.

We apologize for this error. The text should have referred to Table 1 (corrected Line 234 and 247 of the revised manuscript).

10. Ln 308: insert "heavy-isotope labeled" before "minimal medium".

The suggestion has been incorporated into the revised manuscript. The requested modification has been inserted Line 291 of the revised manuscript

11. Ln 467 "Tri-Tri and Tetra-Tri dimers formed by PBPs and YcbB, respectively" should be changed to "Tri-Tri and Tetra-Tri dimers formed by YcbB and PBPs, respectively".

We apologize for this error and have corrected the text accordingly (Line 425 of the revised manuscript).

12. Ln 468: "were almost exclusively of the new-old type (Figure 7)" But Figure 7 A (Tri-Tri) and B (Tetra-Tri) show an even mixture of both new-old and new-new. Hence, the statement "exclusively of new-old" is not entirely correct.

We respectfully disagree with the comment of the reviewer. The proportion of the new→new isotopologue remained close to 0% in the first 20 min for the Tri→Tri (panel A) and Tetra→Tri (panel B) dimers whereas that of the new→old reached 18% during the same time period. The first version of the text may have been misleading since we did not mention that we were considering early times after the medium switch. The corresponding modifications appear in lines 426 to 427 of the revised manuscript: “…. at early times after the medium switch (i.e. ≤ 20 min) (Figure 7A and 7B).”

13. Ln 468-469: "In contrast, Tri-Tetra and Tetra-Tetra dimers formed by these enzymes were a mixture of new-new and new-old" but the figure shows only new-new.

Again, the text was correct in referring to a mixture of new→new and new→old isotopologues for the Tri→Tetra (panel C) and Tetra→Tetra (panel D) dimers at early times.

To improve clarity, we have modified the text to make clear that we are considering short incubation times after the medium switch. In addition, references to specific panels of Figure 7 were added. We also indicated that incubation for longer time periods resulted in an increase in the proportion of new→new isotopologues for all types of dimers and that this increase was expected since serial insertions of glycan strands at the same location results in the formation of new→new dimers.

These modifications appear in lines 427 to 438 of the revised manuscript: “These dimers contained a tripeptide stem at the acceptor position. In contrast, Tri→Tetra and Tetra→Tetra (formed by YcbB and PBPs; Figure 7C and 7D respectively), which contained a tetrapeptide stem at the acceptor position, were mixtures of new→new and new→old types of dimers at early times after the medium switch. Incubation for longer time periods resulted in an increase in the proportion of new→new isotopologues for all types dimers. This increase was expected since serial and independent insertions of glycan strands at the same location results in the formation of new→new dimers. Note that this increase was more rapid for dimers containing a tetrapeptide stem at the acceptor position as observed for tetrapeptide-containing monomers. As described above, this difference reflects the delayed formation of tripeptide stems from tetrapeptide stems by cleavage of D-Ala4 by the L,D-carboxypeptidase activity of YcbB. At early times, tripeptide-containing new stems may therefore be present in small amounts.”

14. Ln 475: The statement "Thus, the mode of insertion of PG subunits depended upon the structure of the acceptor stem … rather than upon the type of transpeptidase" is based on the observation from Figure 7 which shows that panels A and B (Tri as acceptor), and panels C and D (Tetra as acceptor) showed similar kinetic behaviors. The addition of a few sentences for an explanation would add clarity to the statement.

We have rewritten the conclusion of this section and extended it with an explanation (lines 439 to 443 of the revised version):

“Together, the results shown in Figure 7 revealed that the kinetics of formation of isotopologues containing a tripeptide stem at the acceptor position were similar for the Tri→Tri and Tetra→Tri dimers generated by YcbB and PBPs, respectively. Likewise, the kinetics of formation of dimers containing a tetrapeptide stem at the acceptor position were similar for the Tri→Tetra and Tetra→Tetra generated by YcbB and PBPs, respectively.”

15. Figure 7 provides insight into the flux of how much new (unlabeled) PG is incorporated. For example, in Figure 7 C (Tri-Tetra 3-3 CL) at 85 mins the observed N-N is at ~80%, O-O at ~10%, and N-O at ~10%, then the relative total amount of Old PG subunit (2*10% + 10% = 30%) to the New PG subunit (2*80% + 10% = 170%) for Tri-Tetra of 3-3 CL is 3:17. The approximate ratio of old to new for Tetra-Tetra of 4-3 is approximately 1:19. This indicates that the newly formed PG subunits (nascent PG) are incorporated into both 4-3 and 3-3 CL with a tetrapeptide stem, although more actively to 4-3 CL. In contrast, the approximate ratio of old to new for Tri-Tri (3-3 CL) and Tetra-Tri (4-3) is 9:17 and 1:2, respectively. This indicates that the newly formed PG subunit in the form of a tripeptide acceptor is not readily incorporated into the cell wall. Hence, the insertion into PG shows dependence on the PG stem structure of the acceptor. Alternatively, the concentrations of Tri and Tetra are not independent but coupled, most likely high concentration of Tetrapeptide PG stems (both monomers and dimers) as shown Figure 6 (b-f) is likely to be the determinant for the observed kinetics.

We thank the reviewer for these suggestions and have provided a more detailed analysis of the data that takes them into account (lines 443 to 451 of the revised manuscript):

“In contrast, distinct kinetics were observed between dimers containing a stem tripeptide or tetrapeptide in the acceptor position irrespective of the transpeptidase responsible for their formation (mainly new→old versus a mixture of new→new and new→old). Thus, the mode of insertion of peptidoglycan subunits depended upon the structure of the acceptor stem (tripeptide versus tetrapeptide) rather than upon the type of transpeptidase (PBPs *versus* YcbB). In addition, it is worth noting that the newly formed peptidoglycan subunits are more rapidly incorporated into dimers containing a tetrapeptide stem in the acceptor position than into dimers containing a tripeptide stem at this position.”

16. Ln 519: Please explain the amounts of antibiotics that were added, Mec (2.5 ug/ml) and AZT (12 ug/ml), to *E. coli* before the medium switch.

We used inhibitory concentrations of the drugs as mentioned Lines 481 to 483 of the revised manuscript:

“Aztreonam and mecillinam and were used at inhibitory concentrations corresponding to 200- and 20-fold the minimal inhibitory concentration (MIC) of the drugs.

17. Ln 557 Figure 9A: "… grown in the presence of ampicillin", at what time point was the sample collected?

The following text was appended to the sentence (Lines 1084 and 1085 of the revised manuscript):

“….. which was added 5 min prior to the medium switch and to the unlabeled culture medium used for the medium switch corresponding to t=0.”

18. Ln 546-549: "New-new rather than new-old isotopologues were predominantly detected among dimers……. In contrast, the Tri-Tetra dimers in the PG of M1.5 grown in the absence of ampicillin mainly contained the new-old isotopologues (Figure 7)". Figures below show kinetics for Tri-Tetra dimers from Figure 9 (left) and Figure 7C (right). Contrary to the statement, PG of M1.5 grown in the absence of ampicillin (Figure 7C, right) contained a significant amount of new-new isotopologues.

In Figure 7C, the curve for the new→old isotopologues (grey curve) was above that for the new→new isotopologue (purple curve) during the first 20 min of the reaction (for the Tri→Tetra dimer). This indicates that the new-old isotopologues were more abundant than the new-new isotopologues for the Tri→Tetra dimers from the peptidoglycan of M1.5 grown in the absence of ampicillin. We rephrased the sentence to indicate this fact and avoid the use of the word “predominantly” and to indicate that the first 20 min of the reaction were taken into account (Lines 501 to 503):

“In contrast, the new-old isotopologues were more abundant than the new-new isotopologues in the Tri→Tetra dimers from the peptidoglycan of M1.5 grown in the absence of ampicillin at early times (Figure 7).”

19. For Figures 5, 7, 8, and 9, please provide which program was used to fit the data. Also, what was the kinetic model used for the fit? Does the model assume sequential event of A to B to C (conversion from old-old to new-old and then to new-new)? An explanation for a buildup of new-old in Tri-Tri but not in Tri-Tetra in *E. coli* M1.5 grown in the presence of ampicillin (as shown in Figure 9C below) would be helpful.

The curves are interpolations. This is indicated in the revised legends to the figures. We do not want to propose particular models since there are too many variables (See Figure 1 – figure supplement 3).

20. Ln 577-579: The rationale for switching the medium from labeled to unlabeled was that it "facilitates the detection of neo-synthesized isotopologues present at a low abundance at early times after the medium switch." It was not clear how this facilitates the detection of neo-synthesized isotopologues compared to switching from unlabeled to label medium. Additional sentences explaining this would be of interest to the readers.

The additional sentences appear lines 536 to 542 of the revised manuscript.

21. Ln 610: "old-new" should be changed to "old-old".

We added the old→old isotopologues in the sentence. Please note that the procedure is expected to enable the detection of old→new isotopologues although they were not present in the peptidoglycan samples examined in the current study. Accordingly, the reference to old→new isotopologues was maintained in the sentence (Lines 570 and 571 of the revised manuscript).

22. Ln 671-673: Confusing statement. Ln 672: "new-old" should be "new-new"?

We apologize for this confusing statement. The sentence was ambiguous and has been rephrased, Lines 632 to 634 of the revised manuscript:

The resulting new→new cross-linked material was almost exclusively connected to the existing peptidoglycan by the formation of Tri→Tri dimers because new→old isotopologues were only detected in this type of dimers.

23. Figure S12 is of great significance and thus recommend moving this as one of the main figures in the text.

According to the reviewer’s recommendation, Figure S12 appears in the revised version of the main text as Figure 11.

24. Page 20 "Kinetic analyses revealed that the Tri-Tri and Tetra-Tri dimers formed by PBPs and YcbB, respectively, were almost exclusively of the new-old type (Figure 7). Should these be YcbB and PBPs, respectively.

The text has been corrected as suggested (Line 425 of the revised manuscript).

25. In Figure 6, S7, S8 and S9 strain M1.4 is used, but this strain is not described in the method section.

We apologize for these errors. The mutant is M1.5 (not M1.4).

26. The *E. coli* BW25113 rph mutation is a frame shift that leads to pyrimidine starvation. To have an optimal DNA replication cycle in M9 medium it needs some supplements, which were not added according to the Materials and methods. This may not affect the PG mode of insertion but perhaps the authors can keep this in mind for future experiments? All strains used in this study are derivatives of this strain. *E. coli* BW25113 [Δ(araD-araB)567 Δ(rhaD-rhaB)568 ΔlacZ4787 (::rrnB-3) hsdR514 rph-1, lacI+] BW25113 Needs uracil 20 ug/ml, when grown in minimal medium. KF Jensen J Bacteriol. 1993 Jun;175(11):3401-7. Grenier, F., Matteau, D., Baby, V., & Rodrigue, S. (2014). Complete Genome Sequence of *Escherichia coli* BW25113. Genome Announcements, 2(5). doi:10.1128/genomeA.01038-14. Needs also thymidine, 2 ug/ml according to this paper. J Bacteriol. 1998 Jun;180(11):2992-4. Determining the optimal thymidine concentration for growing Thy- Escherichia coli strains. Molina F1, Jiménez-Sánchez A, Guzmán EC.

We thank the reviewer for this information and will further explore the requirements of strain BW25113 for growth. In our hands, BW25513 does not require the addition of uracil or thymidine to the minimal medium. Genome sequencing did not reveal any mutation in our laboratory subculture of BW25113 in comparison to the reference genome. We also found a publication that questions the growth requirement of MG1655, which also harbors the *rph* frameshift mutation “The growth medium did not contain uracil, which has been shown to stimulate growth of *E. coli* MG1655, which has an *rph* frameshift mutation (Jensen, 1993). However, inclusion of uracil had no effect on logarithmic growth, growth arrest caused by isoleucine starvation, or rescue of growth by addition of isoleucine (data not shown).” (Traxler *et al.* 2008. 68:1128-1148; DOI: 10.1111/j.1365-2958.2008.06229.x)